# PLoP: Precise LoRA Placement for Efficient Finetuning of Large Models

**Soufiane Hayou**[*]
Simons Institute
UC Berkeley

**Nikhil Ghosh**
Flatiron Institute

**Bin Yu**
Dept of Statistics
UC Berkeley

## Abstract

Low-Rank Adaptation is a widely used finetuning method for large models. Its small memory footprint allows practitioners to adapt large models to specific tasks at a fraction of the cost of full finetuning. Different modifications have been proposed to enhance its efficiency by, for example, setting the learning rate, the rank, and the initialization. Another improvement axis is adapter placement strategy: when using LoRA, practitioners usually pick *module types* to adapt with LoRA, such as Query and Key modules. Few works have studied the problem of adapter placement, with nonconclusive results: original LoRA paper suggested placing adapters in attention modules, while other works suggested placing them in the MLP modules. Through an intuitive theoretical analysis, we introduce PLoP (**P**recise **LoRA** **P**lacement), a lightweight method that allows automatic identification of module types where LoRA adapters should be placed, given a pretrained model and a finetuning task. We demonstrate that PLoP consistently outperforms, and in the worst case competes, with commonly used placement strategies through comprehensive experiments on supervised finetuning and reinforcement learning for reasoning.

## 1 Introduction

Low-Rank Adaptation (LoRA, Hu et al. (2022)) is a widely used parameter-efficient fine-tuning (PEFT) methods for large language and vision models. LoRA significantly reduces the computational and memory requirements of finetuning by freezing the pretrained model weights and inserting low-rank matrices into the model. This approach has enabled the adaptation of production-scale models on limited hardware resources while achieving performance comparable to full finetuning.

**LoRA improvements.** Several works have considered improving LoRA performance by e.g. using different learning rates for LoRA modules (Hayou et al., 2024a), using normalized updates (Liu et al., 2024), setting adaptive LoRA rank (Kim et al., 2024; Lu et al., 2025), improving initialization (Hayou et al., 2024b), and many other variants, e.g. (Zhang et al., 2023a; Dettmers et al., 2023; Kopiczko et al., 2024; Zhang et al., 2023b; Tian et al., 2024; Jiang et al., 2024). A critical aspect of LoRA is module selection - deciding which specific components of the model should receive the low-rank adaptation. In practice, instead of selecting individual modules, one selects module types such as "q_proj" (Query modules), "v_proj" (Value modules), etc. In Hu et al. (2022), the authors suggested that inserting LoRA in attention modules (Query, Key, and Value) generally yields the best performance among other possible placements. However, in a recent note (Fomenko et al., 2024), the same authors further explained the difficulty encountered in LoRA adapter placement, and mentioned that optimal placement depends on pretrained model and the finetuning task. Another work He et al. (2021) found that for some models, placing LoRA adapters in MLP modules gives better performance. Faced with this confusion, practitioners generally follow one of these guidelines or insert LoRA adapters in all modules which comes at a higher finetuning cost. Therefore, it is natural to ask:

*Given a model and a task, how can we select target module types for LoRA at a reasonable cost?*

---

[*]Work partially done at Johns Hopkins University. Corresponding author: `hayou@jhu.edu`

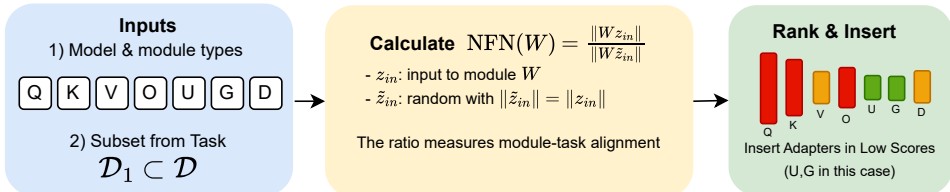

Figure 1: Mechanism of PLoP. For each module $W$ in the model, we calculate alignment scores called NFN (Normalized Feature Norms), rank them, and pick module types with the lowest alignment scores for LoRA insertion. The NFN scores are defined by the ratio between actual activation norms $\|W z_{in}\|$ and a random baseline $\|W \tilde{z}_{in}\|$, where $\tilde{z}_{in}$ is a randomly generated vector with the same euclidean norm as $z_{in}$.

**Memory footprint of LoRA.** In practice, LoRA is used to finetune large models with relatively low cost. Consider Llama3.2-3B (Llama-Team, 2024), processing sequences of 2048 tokens with a batch size of 8. With full finetuning, the memory requirements are substantial. The model parameters require 12GB in float32, while the Adam optimizer states add another 24GB. The activations for a single forward pass consume approximately 48GB of memory. This brings the total memory requirement to approximately 84GB necessitating high-end GPUs. This becomes more problematic with larger, production-scale models. With LoRA, the computational cost changes dramatically. Using rank-16 adapters on query and value modules introduces only 10 million trainable parameters (0.33% of the model). Notably, since gradients are only computed for the adapter weights, the memory overhead for gradient computation is reduced by over 99%. This enables finetuning on a single 24GB GPU with the same batch size and sequence length. These low memory footprint is what makes LoRA attractive for finetuning.

**Anatomy of a practical module selection method for LoRA finetuning.** Based on the computational constraints outlined above, any *practical module selection method for LoRA adapter placement must operate within these resource limitations*. We identify three main pillars of a practical method: (i) the method cannot require computing gradients with respect to the full model parameters, as this would defeat the primary purpose of using LoRA, (ii) the selection process should not necessitate multiple forward passes through different model configurations, as this would multiply the already significant activation memory requirements by the number of candidate configurations being evaluated, (iii) the method must avoid storing large intermediate computations or maintaining extensive state across different module evaluations, which would further strain memory resources. Only methods satisfying these stringent requirements can truly serve practitioners operating in the resource-constrained environments where LoRA provides its greatest value.

In this paper, we introduce PLoP (**P**recise **Lo**RA **P**lacement), a lightweight module placement method for LoRA based on a specific measure of module-data alignment that can be calculated with few forward passes (no gradients, no extensive forward passes, and no storage of intermediate calculations), and therefore, it checks all the three points above (see the compute cost paragraph in Section 3 for more details). The mechanism of PLoP is described in Fig. 1. Specifically, our contributions are as follows:

1. We develop a theoretical framework to study module-data alignment in large neural networks, the core concept behind PLoP.

2. Based on our theoretical analysis of module-data alignment, we develop PLoP, which identifies which module types should be used for LoRA finetuning.

3. We validate our results with extensive experiments showing the benefits of PLoP with LoRA in three post-training scenarios: supervised finetuning for classification, supervised finetuning for text generation, and reinforcement learning for mathematical reasoning.

The paper is structured as follows: In Section 2, we introduce the main theoretical intuition behind our method. In Section 3, we present our method PLoP and provide a quantitative and qualitative analysis of our method. In Section 4, we report empirical results showing the benefit of PLoP in two post-training scenarios: supervised finetuning and reinforcement learning.

## 1.1 RELATED WORK

The effectiveness of LoRA critically depends on the placement of adapter modules. Initially, Hu et al. (2022) studied the placement of adapters in attention modules, observing strong performance in various NLP tasks. He et al. (2021) showed that adapters placed in MLP modules can sometimes outperform attention-based placements. Fomenko et al. (2024) mentioned that optimal adapter placement varies significantly depending on the pretrained model architecture and the downstream task. The authors recommended the following general strategy for adapter placement: start with attention layers, then embeddings, then MLP blocks, and if further capacity is required, raise the LoRA rank. In machine translation, Gheini et al. (2021) found that tuning exclusively cross-attention parameters could achieve performance comparable to full-model tuning.

More adaptive approaches include sensitivity-based parameter selection methods. Zhang et al. (2024) proposed a gradient-based scoring approach that ranks parameters according to their importance to the task, tuning only the highest scoring subset. Similarly, He et al. (2023) developed a sensitivity-aware fine-tuning technique for vision models that dynamically assigns tunable parameters to layers based on local responsiveness. However, such methods require calculating and storing gradients of the full model which is suboptimal for LoRA finetuning (see discussion above). Another variant of LoRA Zhang et al. (2023a) introduces modifications to the adapter structure to adaptively distribute capacity between modules. However, our focus in this paper is on module type selection for LoRA. In our experiments, we compare with two baselines: Insertion in attention modules as recommended by Hu et al. (2022), and Insertion in MLP modules as recommended by He et al. (2021).

Finally, our method is based on a module-data alignment score. Several alignment scores exist in the literature. For instance, Baratin et al. (2021) introduced the centered tangent kernel alignment as a measure of how well aligned each layer is with the task, and Lou et al. (2022) provided a theoretical analysis of such alignment. He et al. (2024) studied the emergence of large feature norms in the network as a result of different training configurations. Our work introduces a new alignment metric based on feature norms.

## 2 FEATURE NORMS CAPTURE MODULE-DATA ALIGNMENT

We provide an intuitive theoretical analysis that shows how feature norms capture information about module-data alignment. Consider a general neural network of the form

$$\begin{cases} Y_{in}(x) = W_{in}x, \\ Y_l(x) = \mathcal{F}_l(W_l, Y_{l-1}(x)), \ l \in [L], \\ Y_{out}(x) = W_{out}Y_L(x), \end{cases} \tag{1}$$

where $x \in \mathbb{R}^d$ is the input, $L \geq 1$ is the network depth, $(\mathcal{F}_l)_{l \in [L]}$ are mappings that define the layers, $W_l \in \mathbb{R}^{n \times n}$ are the hidden weights, where $n$ is the network *width*, and $W_{in}, W_{out}$ are input and output embedding weights.

**Module-Data Alignment.** Fix a training sample $x$. To understand how modules align with the the training sample $x$, we track how hidden features change as we train the model on the singleton $\{x\}$. For the sake of simplification, we consider the case where only a single module $W$ is trained and other modules are frozen. The trainable module has the form

$$z_{out} = W z_{in},$$

where $z_{in} \in \mathbb{R}^n$ is the input, and $z_{out} \in \mathbb{R}^n$ is the output that we call *feature*, both evaluated at the training sample $x$.[1] For Transformer models, the module can be for instance a single Query head, a Projection module in some MLP, etc. The gradient of the loss with respect to the weight matrix $W$ is given by $dW = dz_{out} \otimes z_{in}$, where $dz_{out} = \nabla_{z_{out}} \ell$, the gradient of the loss $\ell$ with respect to feature $z_{out}$, evaluated for the sample $x$. The weights are updated with Adam (Kingma and Ba, 2017), which

---

[1]Here we consider that $z_{in}$ and $z_{out}$ have the same dimension $n$. However, our analysis can be extended to the case where they have different dimensions.

normalizes gradients. Considering the momentum-less version of Adam,[2] updates are given by

$$W_{t+1}z_{in} = W_t z_{in} - \alpha \times \mathcal{S}(dz_{out}^t \otimes z_{in})z_{in}$$
$$= W_t z_{in} - \alpha \times \|z_{in}\|_1 \, \mathcal{S}(dz_{out}^t), \qquad (2)$$

where the superscript in $z_{out}^t = W_t z_{in}$ refers to update step $t$,[3] and $\mathcal{S}$ refers to the sign function (+1 for positive, −1 for negative). In Eq. (5), we used the fact that sign function $\mathcal{S}(.)$ can be expanded across outer product. This is one of the main observations behind the development of $\mu P$ (Yang and Hu, 2022), which sets scaling exponents for initialization and learning rate with respect to model width $n$. We follow $\mu P$ parametrization of the learning rate and set $\alpha = \eta n^{-1}$ for some constant $\eta > 0$ (see Appendix B for more details about the mechanisms of $\mu P$). This yields

$$n^{-1}\|W_{t+1}z_{in}\|_2^2 = n^{-1}\|W_t z_{in}\|_2^2 + \eta^2 n^{-2}\|z_{in}\|_1^2 - 2\eta n^{-1}\|z_{in}\|_1 \times n^{-1}\langle W_t z_{in}, \mathcal{S}(dz_{out}^t)\rangle.$$

The term $n^{-1}\langle W_t z_{in}, \mathcal{S}(dz_{out}^t)\rangle$ measures the alignment between the features $z_{out}^t = W_t z_{in}$ and the "signed" gradients $\mathcal{S}(dz_{out})$. When these two terms are uncorrelated (e.g. at the initial training stages), we have

$$\langle W_t z_{in}, \mathcal{S}(dz_{out})\rangle \approx \mathcal{O}(n^{1/2}), \qquad (3)$$

which yields

$$n^{-1}\|W_{t+1}z_{in}\|_2^2 \approx n^{-1}\|W_t z_{in}\|_2^2 + \alpha^2 n^{-2}\|z_{in}\|_1^2 + \mathcal{O}(n^{-1/2}).$$

Since $\alpha^2 n^{-2}\|z_{in}\|_1^2$ is positive and does not vanish asymptotically (because $\|z_{in}\|_1$ is of size $n$), then the feature norm $n^{-1}\|W_t z_{in}\|_2^2$ increase as training progresses. An in-depth analysis of this phenomenon is provided in Appendix A.

The increase in feature norms indicate *increased alignment between $W$ and $z_{in}$*. To verify this phenomenon in a controlled setting, we consider a three layers linear neural network given by $f(x) = W_2 W_1 W_0 x$, where $x \in \mathbb{R}^d$, $W_0 \in \mathbb{R}^{n \times d}$, $W_1 \in \mathbb{R}^{n \times n}$, and $W_2 \in \mathbb{R}^{1 \times n}$. The training data consist of $N = 1000$ datapoints of dimension $d$ generated from a linear model $y = \omega^\top x + \varepsilon$ with $\varepsilon \sim \mathcal{N}(0, 0.025)$, $\omega_i \sim d^{-1}\mathcal{N}(0, 1)$, and $x$ are generated randomly as standard Gaussian random variables. We use $n = d = 100$ in our experiments and train the model with Adam. See Appendix D for more details and results.

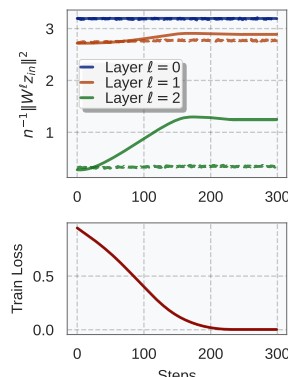

Figure 2 shows the increasing alignment pattern as measured by feature norms for the three layers as we train the model. We include a baseline with no alignment (dashed lines) which shows the norms $\|W\tilde{z}_{in}\|$ where $\tilde{z}_{in}$ is a random Gaussian vector with iid coordinates, normalized such that $\|\tilde{z}_{in}\| = \|z_{in}\|$. The baseline does not show any significant growth over the course of the training which further confirms that feature norms grow as a result of increasing alignment between module weights and module inputs.

Figure 2: Evolution of feature norms during training for the linear network described in Appendix A.1. We train the model for 300 steps with Adam. Feature norms for different layers exhibit differential growth patterns as we train the model. We shifted the curves corresponding to different layers for better visualization.

**Different alignment levels for different modules.** Although we use the same learning rate for all modules, feature norms in the second layer ($n^{-1}\|W_2 z_{in_2}\|^2$) grow much more significantly than those in the input layer ($n^{-1}\|W_0 z_{in_0}\|^2$). This indicates different alignment levels for each module. Such varying alignment patterns between layers has been discussed in Nam et al. (2024) for a different alignment metric.

**Different alignment levels for different inputs/tasks.** In Fig. 2, we report feature norms during training for the actual training inputs $\|Wz_{in}\|$. When evaluating feature norms $\|Wz_{in}'\|$ for an out-of-distribution input $x'$, the resulting alignment depends on how similar is $x'$ to the training samples. The extreme case where $x'$ is very different from the training samples should result in low to no

---

[2]Also known as SignSGD Bernstein et al. (2018). See Appendix for more details.

[3]Note that we do not use such superscript for $z_{in}$ since it does not change when we update $W$.

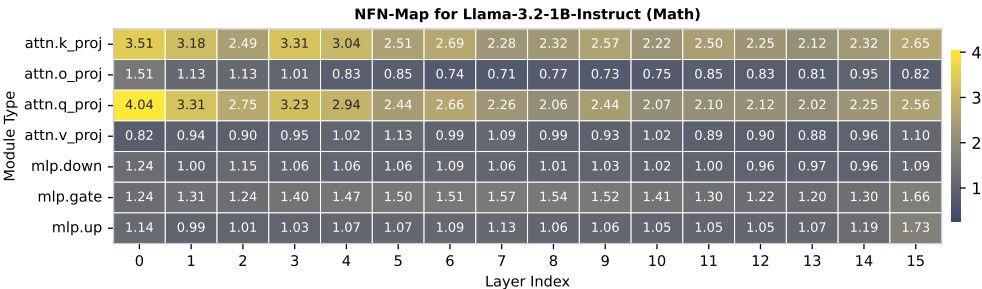

Figure 3: NFN-map for LLama-3.2-1B-Instruct on Math dataset (GSM8K). See Appendix D for NFN-maps of other models.

alignment, as in the random baseline $\|W\tilde{z}_{in}\|$. We provide a more in-depth analysis of this behavior in Appendix B.

This analysis suggest that feature norms can be used to measure module-data alignment in LLMs. In the next section, we refine this notion of alignment and use it to create a method for module type selection for LoRA finetuning.

## 3 PLoP: Finetuning Module Types with the Lowest Alignment

Given a pretrained model and a finetuning dataset $\mathcal{D}$, we compute feature norms for all modules on the task $\mathcal{D}$ by averaging across a subset of $\mathcal{D}$. This captures module-data alignment as detailed in the previous section. Our methodology in based on the following intuition: *The modules with the lowest alignment levels have more potential for adaptation, and therefore should be prioritized in finetuning.*

To obtain a scale-free alignment score, we need to normalize with some baseline. Specifically, as shown in Fig. 2, if we consider a random vector $\tilde{z}_{in}$ with the same norm as $z_{in}$, the feature norm $\|W\tilde{z}_{in}\|$ does not exhibit any significant change throughout training, which is expected due the to lack of correlation between $W$ and $\tilde{z}_{in}$. This random baseline can act as a natural normalizer for the actual feature norms, which leads to the following intuitive definition of our alignment metric.

**Definition 1** (Normalized Feature Norm (NFN)). *Given a pretrained model, a module with weight $W$ in this model, and an input $x$, we define the Normalized Feature Norm as*

$$\text{NFN}(W, x) = \frac{\|W z_{in}(x)\|}{\|W \tilde{z}_{in}(x)\|},$$

*where $\tilde{z}_{in}(x)$ is a vector of the same dimension and norm of $z_{in}(x)$, with i.i.d coordinates distributed as centered Gaussian random variables.*

By incorporating the random baseline $\|W\tilde{z}_{in}(x)\|$, in the definition of NFN scores, we expect to see scores NFN$> 1$ when $W$ is well aligned with $z_{in}$, while the NFN score should be $\approx 1$ when alignment is not significant. Under some assumptions on $W$ and $z_{in}$, we can prove that when the width is large enough, the NFN score can be approximated by $\text{NFN}(W, x) \approx \left\|\frac{W z_{in}(x)}{\|W\|_F \|z_{in}(x)\|}\right\|$ where $\|W\|_F = \sqrt{\sum_{ij} W_{ij}^2}$ is the Frobenius norm of $W$. This approximation shows that dividing by $\|W\tilde{z}_{in}\|$ essentially normalizes $W$ and $z_{in}$. While both forms are cheap to compute, we prefer the expression in Definition 1 as it provides an intuitive interpretation of what NFN scores are representing.

From this analysis and the intuition above, we introduce PLoP, a method that leverages NFN scores to identify which modules should be prioritized for LoRA finetuning. Our method is described below.

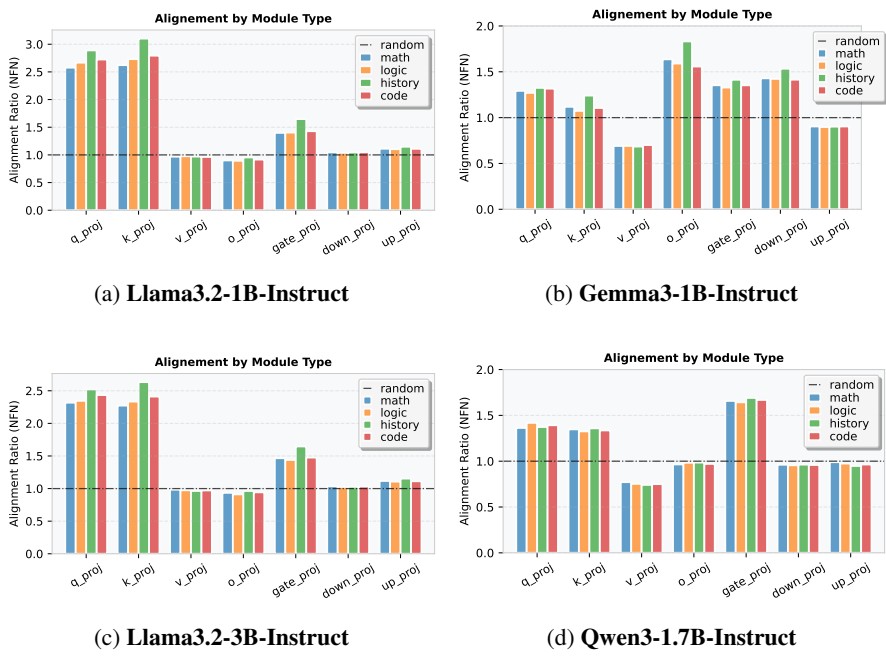

Figure 4: NFN scores aggregated by module type for different models. The scores are for different datasets (math, code, history, and logic).

> **PLoP — Module Type Selection**
>
> **Inputs**: Model $\mathcal{M}$, Finetuning dataset $\mathcal{D}$.
> **Step1 (Scores)**: For each model type $T \in \{$Query, Key, Value, OutProj, GateProj, UpProj, DownProj$\}$, compute average NFN score across $W \in T$ and $x \in \mathcal{D}$.
> **Step2 (Insertion)**: Insert LoRA in module types with the lowest NFN scores.

As stated above, PLoP is based on the hypothesis that modules with the lowest alignment have higher potential for adaptation, and thus should be targeted in finetuning.

For ablation analysis, we also experiment with the reverse PLoP method where instead of choosing module types with the lowest scores, we choose the ones with the highest scores. We call this method PLoP$^{-1}$ and we evaluate its performance in Section 4.

Figure 3 and Fig. 4 show NFN scores for for different models and 4 tasks: *math* (GSM8K, Cobbe et al. (2021)), *code* (HumanEval, Chen et al. (2021)), *history* (MMLU high school european history, Hendrycks et al. (2021)), and *logic* (MMLU logical fallacies). The NFN-map in Fig. 3 provides the most granular level of scoring and shows the NFN scores by module. We can see that key and query attention modules are most aligned with the task in this case, while MLP modules are less aligned, suggesting the need for adaptation in those modules. To see this by module type, we aggregate by averaging over all modules of the same type (step 2 in PLoP) and show the results in the Fig. 4 for different models. We observe significant variability of NFN scores across models, module types, and datasets. For Llama3.2-1B, module types with the highest scores (Query, Key) average around 2-3X the baseline ($\approx 1$), and the lowest scores (Value, Gate, Down, Up) hovering around the baseline score of 1. In this case, PLoP indicates that adaptation should be focused on the (value, gate, down, up) modules rather than the attention query and key matrices. Note that this coincides with the recommendation of empirical work by He et al. (2021) for Llama models but is contradictory to the recommendations of Hu et al. (2022) to finetune mainly attention modules.

Qwen3-1.7B shows high alignment in Query, Key, and Gate modules, with lower alignment for other MLP modules, and a low alignment for the Value module ($\approx 0.75$). This indicates the Value modules in Qwen3-1.7 are "negatively" aligned with all datasets, suggesting that inputs to the Value modules are aligned with the smallest singular directions of the Value weight matrices. The same pattern can

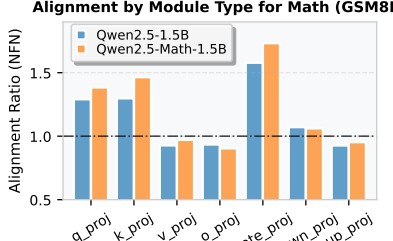 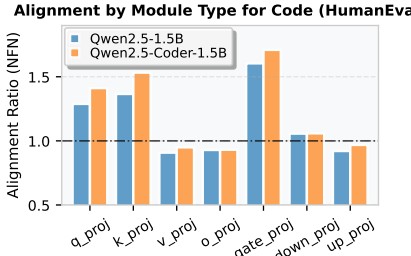

Figure 5: Module types NFN scores for general and specialized Qwen2.5 models. Specialized models (math, code) are finetuned on task-specific data. Scores are higher with the specialized models.

be observed in Gemma3-1B, and we currently do not have an explanation for this phenomenon. In Appendix D, we provide additional NFN scores for Qwen, Gemma, and Llama models.

**NFN scores are task-sensitive.** The alignment scores differ between tasks. For instance, model weights show larger alignment with history compared to math, suggesting that their training data consisted more of sequences similar to general natural language than math related tokens, which is expected. However, note that all tasks share some "base" alignment level given by the general magnitude of the NFN score for each module type. This is a more fundamental phenomenon that is independent of the task, and is related to some basic level of feature learning that is required for token processing. [4]

**NFN scores consistent across different model sizes.** In Fig. 4 (a) and (c), we show NFN scores for two model sizes of Llama3.2, 1B and 3B. The ranking of module types based on NFN scores is roughly the same for both models, suggesting consistency of NFN scores across different model sizes. Intuitively, having similar NFN score patterns suggests similar pretraining and post-training processes for these models, which is expected for models of the same family (Llama3.2 in this case).

**Specialized models show higher NFN scores.** In Fig. 5, we compare NFN scores for instruction-tuned and more specialized version of the same model Qwen2.5-1.5B for math/code tasks. As expected, the specialized models show higher NFN scores overall which further confirms that NFN scores, while cheap to calculate, can be a reliable metric for module-data alignment. The increased NFN score confirm our theoretical analysis in Section 2 and shows that training on domain-specific data increases module-data alignment, captured by the NFN scores.

**Compute cost of** PLoP**.** To obtain the results in Fig. 3, we used a single forward pass with batch size 200, with a maximum sequence length of 256. The cost of PLoP is approximately the same as performing standard batched prefill over 200 inputs, which can be merged with the first step of LoRA finetuning, i.e. roughly zero additional cost. Note that PLoP does not require gradient computation at any stage, making it substantially cheaper than any gradient-based selection strategy.

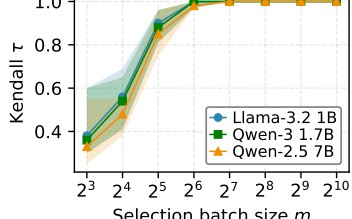

Figure 6: Sensitivity analysis of PLoP in GSM8K dataset.

**Sensitivity of** PLoP **to selection subset size.** As mentioned above, the experiments in Fig. 3, Fig. 4, and Fig. 5 were conducted with a forward propagation batch size of $m = 200$. To understand the sensitivity of PLoP to the choice of $m$, we run the following experiment: for varying batch size $m \in \{2^k, k = 3, \ldots, 10\}$, we divide GSM8K train set (7k+ samples) into pairs of batches of size $m$ (the pair has $2m$ samples in total) and compute the Kendall $\tau$ correlation coefficient between the module types selected by PLoP for each pair. We do this for (dataset size/$2m$) pairs, and show the distribution of Kendall $\tau$ in Fig. 6 as a function of

---

[4]The mechanisms of feature learning in deep neural networks are still largely misunderstood. Quantitative approaches such as Nam et al. (2024) offer some insights, but are far from being comprehensive.

$m$. Across different model sizes, we observe that correlation is close to 1 for batch size $m \geq 64$, indicating that PLoP is also sample efficient. We conduct a similar study for sequence length, which we report in Appendix D.6.

## 4 EXPERIMENTS

We consider three post-training scenarios: Supervised Finetuning for sentence classification, Supervised Finetuning for text generation, and Reinforcement Learning (GRPO, Shao et al. (2024)), all with LoRA adapters. We report results with Llama (Llama-Team, 2024), Qwen (Qwen Team, 2025), and Gemma (Team, 2025) models across different sizes. Our experiments are as follows:

1. SFT for classification: we finetune classifers on ANLI (Nie et al., 2020).
2. SFT for text generation: we consider three experiments: 1) Train on MetaMathQA (Yu et al., 2023) and evaluate the results on GSM8K (Cobbe et al., 2021), 2) Finetune on language datasets from AYA (Singh et al., 2024) and evaluate train/test NLL, and 3) Finetune on CommonsenseQA (Talmor et al., 2019) and evaluate accuracy.
3. RL: we RL-tune on MetaMathQA using GRPO and evaluate on GSM8K.

We investigate the effect of different module placement strategies: our method PLoP(placing LoRA in module types with the lowest NFN scores), Attn (inserting LoRA only in attention modules, as suggested in Hu et al. (2022)), MLP (inserting LoRA only in MLP modules, as suggested in He et al. (2021)), ALL (inserting LoRA in all module types), Random (inserting LoRA in randomly picked module types and average over 4 runs), and as a sanity-check PLoP$^{-1}$ (the inverse of our method, i.e. placing LoRA modules types with the highest NFN scores) to demonstrate that PLoP is capturing meaningful signal.

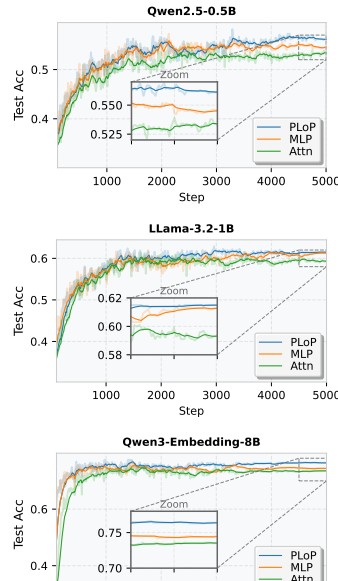

Hereafter, we use the following letters to denote specific modules: **Q** (Query), **K** (Key), **V** (Value), **O** (Out projection), **U** (Up projection), **G** (Gate projection), and **D** (Down projection). All module type NFN scores and experimental details are provided in Appendix D.

### 4.1 SUPERVISED FINETUNING FOR CLASSIFICATION

The Adversarial Natural Language Inference (ANLI) is a language classification task that is more challenging compared to similar tasks (e.g. MNLI). Using LoRA with different placement strategies, we finetune pretrained models on ANLI and report results in Fig. 7. For Qwen2.5-0.5B and Qwen3-Embedding-8B, we observe a significant difference in performance between PLoP and other strategies. For Llama3.2-1B, PLoP and MLP yield roughly the same performance, while Attn is significantly worse. Note that for the Llama model, the MLP modules have small NFN scores, comparable to scores of modules selected by PLoP(V-O-D, see Fig. 4), which could explain why we obtain similar performance with both methods.

Figure 7: LoRA finetuning on ANLI for different models. We use LoRA rank $r = 8$ for MLP strategy and adapt $r$ for PLoP and Attn to match number of parameters for fair comparison. All curves are smoothened with EMA($\alpha = 0.8$) for better visualization. See Appendix D.3 for more details about the experimental setup.

### 4.2 SUPERVISED FINETUNING FOR TEXT GENERATION

**SFT on MetaMathQA.** For challenging tasks such as mathematical reasoning, SFT is often used to "prime" the model for subsequent RL-based finetuning. We perform SFT on the MetaMathQA dataset Yu et al. (2023). To evaluate the performance we measure the accuracy on the GSM8k benchmark Cobbe et al. (2021). All experimental details are provided in Appendix D.3.

We finetune Qwen3 0.6B and 1.7B models. The results are shown in Tables 1 and 2 respectively. In both cases, the placement of adapters in only the attention layers is clearly suboptimal, demonstrating

Table 1: **Qwen3-0.6B**

| Module Types | #Params (M) | Train Loss | Eval Acc |
|---|---|---|---|
| No FT | - | - | 48.1% |
| PLoP(D–U–V) ($r = 76$) | 21.8 | 0.118 | **63.8%** |
| PLoP(D–U–V) ($r = 64$) | 18.4 | 0.116 | 62.0% |
| PLoP$^{-1}$(G–K–Q) ($r = 64$) | 16.5 | 0.119 | 60.6% |
| MLP(D–G–U) ($r = 64$) | 22.0 | 0.119 | 63.3% |
| Attn(K–Q–V) ($r = 64$) | 12.8 | 0.130 | 58.6% |
| all ($r = 64$) | 40.4 | 0.113 | 62.4% |
| Random ($r = 64$) | – | 0.117 | 59.1% |

Table 2: **Qwen3-1.7B**

| Module Types | #Params (M) | Train Loss | Eval Acc |
|---|---|---|---|
| No FT | - | - | 64.7% |
| PLoP(D–O–V) ($r = 102$) | 43.9 | 0.109 | **75.4%** |
| PLoP(D–O–V) ($r = 64$) | 27.5 | 0.111 | 75.2% |
| PLoP$^{-1}$(G–K–Q) ($r = 64$) | 27.5 | 0.113 | 74.6% |
| MLP(D–G–U) ($r = 64$) | 44.0 | 0.108 | 75.0% |
| Attn(K–Q–V) ($r = 64$) | 18.4 | 0.119 | 69.5% |
| all ($r = 64$) | 69.7 | 0.105 | 73.9% |
| Random ($r = 64$) | – | 0.112 | 70.1% |

that for challenging tasks such as mathematics, adapting only the attention layers has limited effect. We see that the most competitive placements for both model sizes is to place adapters according to PLoP or in the MLP layers, even outperforming the placement of adapters in all layers which requires between 1.5-1.8× the number of trainable parameters. When adjusting for an equal number of trainable parameters, PLoP produces the best results with a slight edge over the MLP placement. For the 1.7B parameter model PLoP has a small edge over the MLP placement even with about 60% the number of parameters.

**SFT on Language datasets.** We finetune Llama-3.2 1B and 3B models on languages datasets from the AYA dataset (Singh et al., 2024) with different adapter placement strategies, while matching trainable parameters count. All experimental details are deferred to Appendix D.3. The results are reported in Table 3. On language datasets, PLoP(V-O-D) consistently outperforms other methods, with the second best ranking method is consistently MLP. PLoP$^{-1}$(Q-K-G) shows significant gap in performance, which suggests that picking the modules with the highest NFN scores is suboptimal, and that PLoP is picking meaningful signal. The Attn baseline and Random baseline (average over 4 random selections) rank lower than MLP, confirming previous empirical findings by He et al. (2021), and significantly lower than PLoP.

Table 3: Train/Test NLL on languages from AYA dataset. All methods have comparable trainable parameter count which is fixed by setting $r = 16$ for the MLP method.

| | | Portuguese | | Japanese | | Arabic | |
|---|---|---|---|---|---|---|---|
| | | Train | Test | Train | Test | Train | Test |
| **Llama-3.2 1B** | No FT | – | 7.74 | – | 5.06 | – | 6.82 |
| | Random | 0.475 | 0.491 | 1.81 | 1.95 | 2.13 | 2.33 |
| | Attn | 0.482 | 0.507 | 1.68 | 1.76 | 2.21 | 2.50 |
| | MLP | 0.447 | 0.479 | **1.56** | **1.70** | 0.991 | 1.05 |
| | PLoP$^{-1}$ | 0.512 | 0.612 | 1.73 | 1.91 | 1.79 | 2.33 |
| | PLoP | **0.414** | **0.451** | 1.57 | 1.71 | **0.923** | **0.952** |
| **Llama-3.2 3B** | No FT | – | 7.26 | – | 5.93 | – | 8.72 |
| | Random | 0.356 | 0.421 | 1.44 | 1.50 | 1.59 | 1.87 |
| | Attn | 0.383 | 0.451 | 1.57 | 1.69 | 1.98 | 2.05 |
| | MLP | 0.292 | 0.418 | 1.25 | 1.50 | 0.917 | 0.955 |
| | PLoP$^{-1}$ | 0.356 | 0.431 | 1.61 | 1.79 | 2.01 | 2.13 |
| | PLoP | **0.241** | **0.381** | **1.20** | **1.41** | **0.791** | **0.843** |

**SFT on CommonsenseQA.** We finetune Qwen-2.5 7B and 32B models on CommonsenseQA (Talmor et al., 2019), a dataset consisting of common sense question answering, and evaluate the train NLL and test accuracy. Similar to SFT on language datasets, we consider different adapter placement strategies and match the trainable parameter count for fair comparison. All experimental details are deferred to Appendix D.3. The results are reported in Table 4. The results show that PLoP, which selects V-O-U modules for both model sizes, consistently outperforms other strategies. PLoP$^{-1}$, which selects Q-K-G modules for both module sizes, underperforms other methods indicating that high NFN scores are associated with lower performance.

Table 4: Train NLL/Test Acc on CommonsenseQA. All methods have comparable trainable parameter count which is fixed by setting $r = 8$ for the MLP baseline.

| | | Train NLL | Test Acc |
|---|---|---|---|
| **Qwen-2.5 7B** | No FT | – | 82.8% |
| | Random | 0.258 | 86.2% |
| | Attn | 0.263 | 86.0% |
| | MLP | 0.247 | 87.3% |
| | PLoP$^{-1}$ | 0.264 | 86.7% |
| | PLoP | **0.235** | **88.6%** |
| **Qwen-2.5 32B** | No FT | – | 88.2% |
| | Random | 0.082 | 91.2% |
| | Attn | 0.081 | 91.5% |
| | MLP | 0.075 | 91.9% |
| | PLoP$^{-1}$ | 0.080 | 91.0% |
| | PLoP | **0.072** | **92.6%** |

## 4.3 RLVR FOR MATHEMATICAL REASONING

Reinforcement Learning has emerged as a promising approach for test-time scaling. Algorithms such as GRPO incentivize the model to follow a pattern of "thinking" before providing the final answer. This implicit approach to reasoning (versus the more explicit approaches such as MCTs (Xie et al., 2024)) showed very

promising results, especially with the impressive performance of DeepSeek-R1 (DeepSeek-AI, 2025). In this section, we experimented with "GRPO on a budget" using LoRA adapters (instead of training the full weights) to enhance mathematical reasoning. We select 3 module types for LoRA adapter placement using the strategies stated above. We compare both RL rewards (columns Rwd/Format and Rwd/Answer) and Evals (GSM8K 8shots prompting Pass@1). Note that because LoRA is a lightweight finetuning method, it would not be sufficient to induce reasoning in base models with GRPO, especially with a small rank $r$. For this purpose, we apply GRPO with LoRA to instruction-tuned models instead of base models. For more implementation details, see Appendix D.

Table 5: GRPO results for **Qwen3-1.7B** trained on MetaMathQA.

| Module Types | #Params (M) | Rwd/Format | Rwd/Answer | Eval/GSM8K |
|---|---|---|---|---|
| No GRPO | – | – | – | 65.50% |
| Attn(Q-K-V) ($r = 16$) | 4.58 | 1.89 | 0.91 | 71.49% |
| Attn(Q-K-V) ($r = 25$) | 7.17 | 1.97 | 0.98 | 72.13% |
| MLP(U-G-D) ($r = 16$) | 11.01 | 2.57 | 1.28 | 73.61% |
| PLoP$^{-1}$(Q-K-G) ($r = 16$) | 6.88 | 1.71 | 0.86 | 71.41% |
| PLoP(V-O-D) ($r = 16$) | 6.88 | 2.67 | 1.32 | 74.52% |
| PLoP(V-O-D) ($r = 25$) | 10.75 | **2.75** | 1.32 | **75.03%** |

With GRPO, we define the think-then-answer pattern in similar way to DeepSeek-R1. The model is rewarded for placing the thinking process in between `<think>` and `</think>`, and then giving the answer in between `<solution>` and `</solution>`. This is encoded in the format reward function (Rwd/Format). The correctness of the solution is rewarded as well (Rwd/Answer). We track this reward as we train the model with GRPO and show the final results (at convergence).

Table 5 shows the results of GRPO with the Qwen3-1.7B trained on MetaMathQA. PLoP yields better performance overall both in training rewards and evaluation (GSM8K). Compared to Attn, PLoP performs better even when matching the number of trainable parameters (Attn($r = 25$) vs PLoP($r = 16$)). Interestingly, PLoP performs better than MLP placement strategy even when using the same rank $r = 16$, in which case we have 6.88M trainable parameters with PLoP and 11.01M parameters with MLP. For the same number of parameters, the performance is further improved with PLoP($r = 25$).

We experimented with Gemma3-1B as well, and found that PLoP outperforms other alternatives, although the score on GSM8K is low due the inherent limitations of Gemma3-1B in mathematical reasoning. See Appendix D for more details.

## 5 DISCUSSION AND LIMITATIONS

We introduced PLoP, an intuitive module type selection method, designed specifically for LoRA fine-tuning and based on NFN scores – a notion of module-data alignment supported by an intuitive theoretical analysis. PLoP meets the computational criteria needed for efficient LoRA finetuning as articulated in the introduction: PLoP's lightweight nature makes it particularly valuable in resource-constrained environments where LoRA is most beneficial. PLoP is based on the NFN-map, which enables more granular selection beyond module types. However, we deliberately focused on module type selection as it represents the most widely adopted aggregation approach among practitioners, avoiding the additional implementation complexities of more fine-grained selection. While we explored using PLoP for layer-level selection by inserting LoRA into target layers with low NFN scores, we encountered inconsistent results and have reserved this question for future research.

## 6 ACKNOWLEDGMENT

This work used GPU servers at Delta AI (NCSA) through allocation #CIS250193 from the Advanced Cyberinfrastructure Coordination Ecosystem: Services & Support (ACCESS) program (Boerner et al., 2023), which is supported by U.S. National Science Foundation grants #2138259, #2138286, #2138307, #2137603, and #2138296.

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

# A    THEORETICAL ANALYSIS OF MODULE-DATA ALIGNMENT

Given a pretrained model and a finetuning task, our goal is to strategically place LoRA adapters in modules that would contribute most significantly to performance. In practice, we usually select module types instead of single weight matrices. For instance, for Llama3 models, we might choose to insert LoRA in Query ("q_proj") and Key ("k_proj") modules.

As we discussed in Section 1, for such a method to be useful, it should first be a lightweight method that operates efficiently in resource-constrained environments and ideally rely on existing computation pipelines. Ideally, the method should rely exclusively on standard forward propagation, as this computational pipeline is already necessary for inference and adds no significant overhead to the existing workflow.

This section provide a more in-depth theoretical analysis, complementing that of Section 2.

**Mechanisms behind the growth in feature norms.**    The reason behind the growth in feature norms for certain modules is non-trivial. The naive explanation to this phenomenon is that with training, weight norms grow for some modules and remain constant constant or decrease for others. However, as we will see in the next analysis, the mechanisms behind this phenomenon are more subtle, and the most important factor is a form of alignment that occurs between module weight and its input.

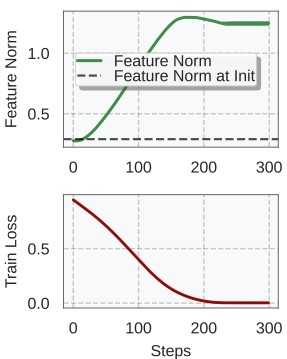

Specifically, we show that this growth in feature norms in some modules appears primarily as a result of two factors: 1) large-width in neural networks (large embedding dimension), a condition that is generally satisfied in practice, [5] and 2) progressive alignment of modules weights with their respective inputs.

Consider a general neural network of the form

Figure 8: Illustration of feature norm growth during training. This shows the feature norms $n^{-1}\|W z_{in}\|^2$ for a module $W$ in the model ($W \in \mathbb{R}^{n \times n}$). See Appendix A.1 for details about the model and training.

$$
\begin{aligned}
Y_{in}(x) &= W_{in}x, \\
Y_l(x) &= \mathcal{F}_l(W_l, Y_{l-1}(x)), \ l \in [L], \\
Y_{out}(x) &= W_{out}Y_L(x),
\end{aligned}
\tag{4}
$$

where $x \in \mathbb{R}^d$ is the input, $L \geq 1$ is the network depth, $(\mathcal{F}_l)_{l \in [L]}$ are mappings that define the layers, $W_l \in \mathbb{R}^{n \times n}$ are the hidden weights, where $n$ is the network *width*, and $W_{in}, W_{out}$ are input and output embedding weights.

Model (4) is pretrained on some data mixture $\mathcal{D}$ to minimize some loss function $\ell$ – the next-token prediction loss in the case of language models. We introduce some notation that will facilitate the presentation of our analysis.

**Notation.**    Hereafter, $n$ will always denote model width. As $n$ grows, given sequences $c_n \in \mathbb{R}$ and $d_n \in \mathbb{R}^+$, we write $c_n = \mathcal{O}(d_n)$ to refer to $c_n < \kappa d_n$ for some constant $\kappa > 0$. We write $c_n = \Theta(d_n)$ if we have $\kappa_1 d_n \leq c_n \leq \kappa_2 d_n$ for some $\kappa_1, \kappa_2 > 0$. For vector sequences $c_n = (c_n^i)_{1 \leq i \leq k} \in \mathbb{R}^k$ (for some $k > 0$), we write $c_n = \mathcal{O}(d_n)$ when $c_n^i = \mathcal{O}(d_n^i)$ for all $i \in [k]$, and same holds for other asymptotic notation. Finally, when the sequence $c_n$ is a vector of random variables, asymptotics are defined in the sense of the second moment ($L_2$ norm). For a vector $z \in \mathbb{R}^n$, we will use the following norms: $\|z\| = \left(\sum_{i=1}^n z_i^2\right)^{1/2}$ (euclidean norm), and $\|z\|_1 = \sum_{i=1}^n |z_i|$ ($\ell_1$ norm).

**Intuitive theoretical analysis.**    For the sake of tractability, we consider the case where a single weight matrix (module) in the model is trained and other modules are frozen. [6] We further simplify

---

[5]From the literature on infinite-width theory, when we take the width to infinity, the training dynamics converge with a rate of roughly $\mathcal{O}(n^{-1/2})$ (Yang and Hu, 2022). In practice, a width of $n \gtrsim 10^3$ is generally considered large enough for the theoretical predictions to be a good approximation of practice.

[6]While this is unrealistic, it provides the right intuition behind our methodology, and makes the analysis more tractable.

the analysis by assuming that the model is trained in a single datapoint $x$. We later discuss the impact of batch training. The trainable module has the form

$$z_{out} = W z_{in},$$

where $z_{in} \in \mathbb{R}^n$ is the input, and $z_{out} \in \mathbb{R}^n$ is the output that we call *feature*, both evaluated at the training datapoint $x$.[7] For Transformer models, the module can be for instance a single Query head, a Projection module in some MLP, etc.

The gradient of the loss with respect to the weight matrix $W$ is given by

$$dW = dz_{out} \otimes z_{in},$$

where $dz_{out} = \nabla_{z_{out}} \ell$, the gradient of the loss with respect to feature $z_{out}$.

In general, modern LLMs are trained with Adam (Kingma and Ba, 2017), which normalizes gradients. In its momentum-less form, Adam becomes SignSGD (Bernstein et al., 2018), which is defined by $\mathcal{S}(dW_{ij}) = +1$ if $dW_{ij} \geq 0$, otherwise $-1$. SignSGD is a nice simplification of Adam: it captures the property of normalization and allows tractable the theoretical analysis as we will see below. With SignSGD, feature updates[8] are given by

$$\begin{aligned} W_{t+1} z_{in} &= W_t z_{in} - \alpha \times \mathcal{S}(dz_{out} \otimes z_{in}) z_{in} \\ &= W_t z_{in} - \alpha \times \|z_{in}\|_1 \, \mathcal{S}(dz_{out}^t), \end{aligned} \tag{5}$$

where the superscript in $z_{out}^t = W_t z_{in}$ refers to update step $t$. Note that we do not use such superscript for $z_{in}$ since it does not change when we update $W$.

The key trick used in Eq. (5) is that the sign function $\mathcal{S}(.)$ can be expanded across outer product. This is one of the main observations behind the development of $\mu$P (Yang and Hu, 2022), which sets scaling exponents for initialization and learning rate with respect to model width $n$. Under $\mu$P, all weights in the model are initialized to have roughly $1/\sqrt{n}$ magnitude (or more precisely $1/\sqrt{fan\_in}$), which implies that features $z_{out}$ and their inputs $z_{in}$ to have $\Theta_n(1)$ norm at initialization (i.e. $n^{-1}\|z_{in}\|_1 = \Theta_n(1)$).

Eq. (5) describes the evolution of features $z_{out}$ as we update weights $W$. Ideally, we want both stability ($W_t z_{in}$ does not grow in magnitude with $n$) and non-triviality ($W_t z_{in}$ does not converge to 0 with $n$). These conditions are both satisfied when $W_{t+1} z_{in} - W_t z_{in} = \Theta_n(1)$ element-wise, which implies that the learning rate should scale as $\alpha = \eta n^{-1}$ for some constant $\eta > 0$, to compensate the growth in $\|z_{in}\|_1$, which is exactly the $\mu$P scaling rule for the learning rate. See Appendix B below for more details about the mechanisms of $\mu$P. With this parametrization of the learning rate, we obtain

$$\|W_{t+1} z_{in}\|_2^2 = \|W_t z_{in}\|_2^2 + \eta^2 n^{-1} \|z_{in}\|_1^2 - 2\eta n^{-1} \|z_{in}\|_1 \times \langle W_t z_{in}, \mathcal{S}(dz_{out}^t) \rangle.$$

We can normalize by $n$ so terms on both sides have $\Theta_n(1)$ magnitude in width $n$,

$$n^{-1}\|W_{t+1} z_{in}\|_2^2 = n^{-1}\|W_t z_{in}\|_2^2 + \eta^2 n^{-2} \|z_{in}\|_1^2 - 2\eta n^{-1} \|z_{in}\|_1 \times n^{-1} \langle W_t z_{in}, \mathcal{S}(dz_{out}^t) \rangle.$$

The term $n^{-1} \langle W_t z_{in}, \mathcal{S}(dz_{out}^t) \rangle$ measures the alignment between the features $z_{out}^t = W_t z_{in}$ and the "signed" gradients $\mathcal{S}(dz_{out})$. Intuitively, at the initial training stages, these two terms are roughly independent (as random variables) because of the randomness from the initialization weights. As a result, in those initial training stages, we have

$$\langle W_t z_{in}, \mathcal{S}(dz_{out}) \rangle \approx \mathcal{O}(n^{1/2}), \tag{6}$$

which yields

$$n^{-1}\|W_{t+1} z_{in}\|_2^2 \approx n^{-1}\|W_t z_{in}\|_2^2 + \alpha^2 n^{-2} \|z_{in}\|_1^2 + \mathcal{O}(n^{-1/2})$$

---

[7]Here we consider that $z_{in}$ and $z_{out}$ have the same dimension $n$. However, our analysis can be extended to the case where they have different dimensions.

[8]Feature update is the change of the features $z_{out}$ after taking one training step.

Since $\alpha^2 n^{-2} \|z_{in}\|_1^2 = \Theta_n(1)$ is positive and asymptotically non-zero, if the width $n$ is large enough, we should expect the (normalized) feature norm $n^{-1}\|W_t z_{in}\|_2^2$ to grow initially during training. The next results provides a rigorous description of this phenomenon for linear networks.

**Theorem 1** (Feature Norm Growth in Linear Networks (Informal)). *Assume that the neural network is a linear MLP (see Appendix B for more details). Then, for any $\delta \in (0, 1/2)$, under some assumptions stated in Appendix C, there exists a universal constant $\lambda > 0$ such that for any $T$ and $\eta$ satisfying $T \le \lambda \eta^{-1}$, the following holds with probability at least $1 - 2n^{-1+2\delta}$*

$$\sup_{1 \le t \le T} \left| n^{-1}\|W_t z_{in}\|^2 - \Gamma_t \right| \le Cn^{-\delta}, \tag{7}$$

*where $\Gamma_t = \Gamma_0 + \beta^2(1 + t(t-1))$, $\beta = \eta\, n^{-1}\|z_{in}\|_1$, and $\Gamma_0 = n^{-1}\|W_0 z_{in}\|^2$. In other words, when the width $n$ is large enough, $n^{-1}\|W_t z_{in}\|^2$ exhibits quasi-quadratic growth at initial training stages.*

Theorem 1 characterizes the growth in feature norms $n^{-1}\|W_t z_{in}\|^2$ as training progresses. The proof is provided in Appendix C. In this case, $n^{-1}\|W_t z_{in}\|^2$ grows in a quasi-quadratic pattern, which becomes perfectly quadratic when $n \to \infty$. This is the most important takeaway from this result: this phenomenon is associated with large width. With more realistic models, we expect the growth property to hold, but not necessarily with the quadratic form. See next section for empirical results.

## A.1 Evolution of Feature Norms

Consider a three layers linear neural network given by $f(x) = W_2 W_1 W_0 x$, where $x \in \mathbb{R}^d$, $W_0 \in \mathbb{R}^{n \times d}$, $W_1 \in \mathbb{R}^{n \times n}$, and $W_2 \in \mathbb{R}^{1 \times n}$. The training data consist of $N = 1000$ datapoints of dimension $d$ generated from a linear model $y = \omega^\top x + \varepsilon$ with $\varepsilon \sim \mathcal{N}(0, 0.025)$, $\omega_i \sim d^{-1}\mathcal{N}(0, 1)$, and $x$ are generated randomly as standard Gaussian random variables. We use $n = d = 100$ in our experiments and train the model with Adam. See Appendix D for results with SignSGD and more details about the experimental setup.

Figure 9 shows the growth in feature norms for the three modules (corresponding to the three layers in this case) as we train the model. We include a baseline (dashed lines) which shows the norms $\|W \tilde{z}_{in}\|$ where $\tilde{z}_{in}$ is a random Gaussian vector with iid coordinates, normalized such that $\|\tilde{z}_{in}\| = \|z_{in}\|$ (see next section for an intuitive explanation of this baseline). The baseline does not show any significant growth over the course of the training which further confirms that feature norms grow as a result of increasing alignment between module weights and module inputs, and not simply as a result of an increase in weight norms.

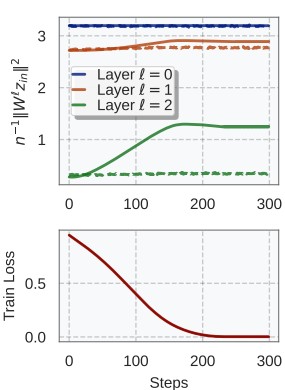

Figure 9: Evolution of feature norms during training for the linear network described in Appendix A.1. We train the model for 300 steps with Adam. Feature norms for different layers exhibit differential growth patterns as we train the model. We shifted the curves corresponding to different layers for better visualization.

**Most of the growth occurs early in training.** Interestingly, most of the growth in feature norms occurs in the first $T = 200$ steps, which also correlates with the most significant drop in training loss. After $T = 200$, feature norms remain roughly stable until convergence. This suggest that the norm growth is associated with an initial phase where significant feature learning occurs, and remains roughly unchanged after that initial growth phase. Intuitively, as we train the network, the dot product between $z_{out}$ and $S(dz_{out})$ (Eq. (6)) grows from $\mathcal{O}(n^{1/2})$ to roughly $\Theta_1(n)$ (in absolute value) and therefore the argument behind the feature norm growth as explained in the discussion above no longer holds later in training. As a result, the growth plateaus after some number of steps $T$.

**Different growth levels for different modules.** Although we use the same learning rate for all modules, the norm growth in the input layer ($n^{-1}\|W_0 z_{in_0}\|^2$) is much less significant than that observed in the second layer ($n^{-1}\|W_2 z_{in_2}\|^2$). To understand this difference, we should take into account that when training all modules (layers in this case), the inputs to $W_1$ and $W_2$ change with training. The update in feature are given by

$$W_{t+1} z_{in}^{t+1} = W_t z_{in}^t - \eta\, n^{-1} \times \|z_{in}^t\|_1\, \mathcal{S}(dz_{out}^t) + W_{t+1}\Delta z_{in}^{t+1},$$

where $\Delta z_{in}^{t+1} = z_{in}^{t+1} - z_{in}^t$ is the input change after one step. Under $\mu$P scaling rule for the learning rate $(\eta\, n^{-1})$, the magnitude of $\|z_{in}^t\|_1$ remains $\Theta_n(1)$ for all $t$, however the constant in $\Theta_n(1)$ naturally depends on the layer. Additionally, the term $W_{t+1}\Delta z_{in}^{t+1}$ introduces more complex update dynamics, and contribute in a non-trivial way to the change in the feature norms. See Nam et al. (2024) for a more detailed discussion on how feature learning changes from one layer to another. Both of these aspects lead to uneven growth in the feature norms for different layers.

**Different growth levels for different inputs/tasks.** In the setup of Theorem 1, we considered a batch size of 1, which results in feature updates of the form $W_{t+1}z_{in} = W_t z_{in} - \eta \times n^{-1}\|z_{in}\|_1\, \mathcal{S}(dz_{out})$, and we saw that with $1/\sqrt{n}$ initialization, we have $n^{-1}\|z_{in}\|_1 = \Theta_n(1)$. In the realistic setting of batch training, feature updates for an input $x' \in \mathbb{R}^d$ are given by

$$W_{t+1}z_{in}(x') = W_t z_{in}(x') - \eta\, n^{-1} \times \mathcal{S}\left(\frac{1}{|B|}\sum_{x \in B} dz_{out}(x) \otimes z_{in}(x)\right) z_{in}(x'),$$

Therefore, we can no longer directly expand the sign function and obtain the $\|z_{in}\|_1$ term that leads to the $\Theta_n(1)$ term. In this case, we need a strong correlation between $\mathcal{S}\left(\frac{1}{|B|}\sum_{x \in B} dz_{out}(x) \otimes z_{in}(x)\right)$ and $z_{in}(x')$ to obtain the same effect. This translates to whether the datapoint $x'$ has some similarity with the batch used for the update. As a result, we should expect to see higher scores for datapoints that are similar to the training dataset, and lower scores for significantly different datapoints.

## B  ADDITIONAL THEORETICAL DETAILS

### B.1  INFINITE-WIDTH ANALYSIS AND $\mu$P

Scaling remains the main paradigm to improve performance of language model (see e.g. Hoffmann et al. (2022)). This includes model capacity which can be increased via width (embedding dimension) or depth (number of layers) or both, training data, number of training steps etc. In our theoretical analysis in Section 2, we mentioned the infinite-width $n \to \infty$ and how our results hold in this limit. This is motivated by the fact that most state-of-the-art language and vision models have large width.

As the width $n$ grows, most hyperparameters in the model such as the initialization and the learning should be adapted to avoid numerical instabilities and ensure efficient learning. For instance, the initialization variance should scale as $1/n$ to prevent arbitrarily large pre-activations as we increase model width $n$ (e.g. He init He et al. (2015)). To derive such scaling rules, a principled approach consist of analyzing statistical properties of key quantities in the model (e.g. pre-activations) as $n$ grows and then adjust the initialization, the learning rate, and the architecture itself to achieve desirable properties in the limit $n \to \infty$ Hayou et al. (2019); Yang (2019).

In this context, Yang and Hu (2022) introduces the Maximal Update Parameterization (or $\mu$P), a set of scaling rules for the initialization scheme, the learning rate, and the network architecture that ensure stability and maximal feature learning in the infinite width limit. Stability is defined by $Y_l^i = \Theta(1)$ for all $l$ and $i$ where the asymptotic notation '$\Theta(.)$' is with respect to width $n$ (see next paragraph for a formal definition), and feature learning is defined by $\Delta Y_l = \Theta(1)$, where $\Delta$ refers to the feature update after taking a gradient step. $\mu$P guarantees that these two conditions are satisfied at any training step $t$. Roughly speaking, $\mu$P specifies that hidden weights should be initialized with $\Theta(n^{-1/2})$ random weights, and weight updates should be of order $\Theta(n^{-1})$. Input weights should be initialized $\Theta(1)$ and the weights update should be $\Theta(1)$ as well. While the output weights should be initialized $\Theta(n^{-1})$ and updated with $\Theta(n^{-1})$. These rules ensure both stability and feature learning in the infinite-width limit, in contrast to standard parameterization (exploding features if the learning rate is well tuned), and kernel parameterizations (e.g. Neural Tangent Kernel parameterization where $\Delta Y_l = \Theta(n^{-1/2})$, i.e. no feature learning in the limit).

## C  PROOF OF THEOREM 1

In this section, we provide the full proof for Theorem 1. Forst, we prove a result on the sign of the derivative of the loss function with respect to $z_{out}$, then proceed with the full proof.

### C.1 CONSTANT LOSS DERIVATIVE SIGN IN THE INITIAL TRAINING STAGE

Consider a linear network of the form

$$f(x) = W_L W_{L-1} \dots W_0 x, \quad x \in \mathbb{R}^d, \tag{8}$$

where $L \geq 1$ is the network depth, $W_\ell \in \mathbb{R}^{n \times n}$ for all $\ell \in \{1, 2, \dots, L-1\}$ are hidden layers parameters, $W_L \in \mathbb{R}^{1 \times n}$ is the projection layer weight, and $W_0 \in \mathbb{R}^{n \times d}$ is the input layer weight.

The network is trained with the setup described in Section 2, namely:

- Dataset: a single datapoint $(\hat{x}, \hat{y}) \times \mathbb{R}^d \times \mathbb{R}$.
- Training algorithm: SignSGD with learning rate $\eta n^{-1}$.
- Single layer: only a single hidden layer that we denote $W \in \{W_\ell, \ell = 1, 2, \dots, L-1\}$ is trained, and other layers weights are fixed to their values at initialization. Without loss of generality, assume that the trainable layer is $\ell_0$, i.e. $W = W_{\ell_0}$.
- Training loss: quadratic loss given by $\mathcal{L}(W) = 2^{-1}(f_W(\hat{x}) - \hat{y})^2$.

In this setting, the linear network training dynamics become tractable and we can obtain closed-form expressions in steps $t$ and width $n$. Let us use the same notation as in Section 2 and denote the input and output of the trainable layer $z_{out} = W z_{in}$. More precisely, in this case, we can express the network output as $f_W(x) = V^\top z_{out} = V^\top W z_{in}$, where $z_{in} = Mx$, with $M = W_{\ell_0-1} \dots W_0 \in \mathbb{R}^{n \times d}$ and $V^\top = W_L \dots W_{\ell_0+1} \in \mathbb{R}^{1 \times n}$ are both non-trainable random matrices. In this case, the gradient of the loss with respect to $z_{out}$ is given by

$$dz_{out} = (V^\top W z_{in} - y)V.$$

From now on, we will abuse the notation and use the subscript to denote the training step as well for the matrix $W = W_{\ell_0}$. When we use the notation $W_t$, it should interpreted as $W_{\ell_0,t}$. Taking one step with SignSGD yields

$$W_{t+1} = W_t - \eta\, n^{-1} \|z_{in}\|_1 \chi_t \mathcal{S}(V),$$

where $\mathcal{S}(.) = sign(.)$ and $\chi_t = \mathcal{S}(V^\top W_t z_{in} - \hat{y})$.

Next, we prove a result that will be useful in the proof of Theorem 1. More precisely, we show that under mild assumptions, there exists a first initial training phase in which the sign of the loss function on the training datapoint does not change. The number of steps in this phase is bounded by $\eta^{-1}$ up to some constant factor. Naturally, since we initialize with random variables, it should be expected that such result could only hold with high probability.

**Theorem 2** (Constant loss derivative sign in the initial training phase)**.** *We assume that the weights $W_0, W_1, \dots, W_L$ are initialized such that the following holds:*

- *$|z_{in}^i| \in [\underline{Z}, \bar{Z}]$ for all $i \in [1:n]$, where $\underline{Z}, \bar{Z} > 0$ are constants independent of $n$.*
- *Mean-field Init: $\mathbb{E}[V_i] = 0$ and $Var(V_i) = n^{-2}$ (e.g. uniform distribution on $[-n^{-1}, n^{-1}]$).*

*Further assume that $y \in [\underline{Z}, \bar{Z}]$.* [9]

*Then, for any $\delta \in (0, 1)$, and $T \leq \eta^{-1}\left(n^{-\delta} + \frac{\bar{Z}}{\underline{Z}^2}\right)$, we have with probability at least $1 - n^{-1+\delta}$,*

$$\forall t \leq T, \chi_t = \chi_0.$$

The assumption on the weight initialization is mild and is satisfied by some standard initialized schemes, such as uniform init with $n^{-1}$ variance for the hidden weights, $d^{-1}$ variance for the input weights, and $n^{-2}$ variance for the projection weights. The proof of Theorem 2 relies on standard concentration results.

---

[9]This can satisfied with a simple adjustment of the constants $\underline{Z}, \bar{Z}$.

*Proof.* Recall the definition of $\chi_t$

$$\chi_t = \mathcal{S}(V^\top W_t z_{in} - \hat{y}).$$

We have the following from above,

$$W_t z_{in} = W_{t-1} z_{in} - \eta n^{-1} \|z_{in}\|_1 \chi_{t-1} \mathcal{S}(V),$$

and therefore,

$$V^\top W_t z_{in} = V^\top W_{t-1} z_{in} - \eta n^{-1} \|z_{in}\|_1 \|V\|_1 \chi_{t-1}$$

which implies that

$$V^\top W_t z_{in} = V^\top W_0 z_{in} - \eta n^{-1} \|z_{in}\|_1 \|V\|_1 \left[ \sum_{j=0}^{t-1} \chi_j \right].$$

**Bounding $V^\top W_0 z_{in}$:**

With Chebyshev's inequality we have:

$$\mathbb{P}\left( |V^\top W_0 z_{in}| \geq \bar{Z} n^{-\delta} \right) < \frac{\mathrm{Var}(V^\top W_0 z_{in})}{(\bar{Z} n^{-\delta})^2}$$

where

$$\mathrm{Var}(V^\top W_0 z_{in}) = \frac{1}{n} \mathrm{Var}(W_0^{i\top} z_{in}) = \frac{1}{n} \cdot \frac{\|z_{in}\|^2}{n} \leq n^{-1} \bar{Z}^2$$

As a result, we obtain:

$$\mathbb{P}\left( |V^\top W_0 z_{in}| \geq \bar{Z} n^{-\delta} \right) < n^{-1+2\delta}$$

If $V^\top W_0 z_{in} - \hat{y} < 0$, we have

$$V^\top W_t z_{in} - \hat{y} \leq V^\top W_0 z_{in} - \hat{y} + \eta n^{-1} \|z_{in}\|_1 T$$
$$\leq V^\top W_0 z_{in} - \hat{y} + \eta \bar{Z} T$$

With probability at least $1 - n^{-1+2\delta}$, we have

$$V^\top W_t z_{in} - \hat{y} \leq \bar{Z} n^{-\delta} - \hat{y} + \eta \bar{Z} T$$

Therefore, we have

$$T \leq \eta^{-1} \left( \frac{\hat{y}}{\bar{Z}} - n^{-\delta} \right) \Rightarrow \forall t \leq T, \quad \chi_t = \chi_0 = -1.$$

If $V^\top W_0 z_{in} - \hat{y} > 0$, asymptotically this implies that $-\hat{y} > 0$ (assuming $|\hat{y}| = \Theta_n(1)$). Similarly, we obtain with probability at least $1 - n^{-1+2\delta}$,

$$V^\top W_t z_{in} - \hat{y} \geq V^\top W_0 z_{in} - \hat{y} - \eta \bar{Z} T,$$
$$\geq -\bar{Z} n^{-\delta} - \hat{y} - \eta \bar{Z} T,$$

and therefore, we have that

$$T \leq \eta^{-1} \left( \frac{-\hat{y}}{\bar{Z}} - n^{-\delta} \right) \Rightarrow \forall t \leq T, \quad \chi_t = \chi_0 = 1.$$

In summary, we have the following: Let $\delta \in (0, 1)$. Then, with $T \leq \eta^{-1} \left( \frac{|\hat{y}|}{\bar{Z}} - n^{-\delta} \right)$, we have for all $t \leq T$, $\chi_t = \chi_0$.

$\square$

The assumptions in Theorem 2 can be alleviated to include more generalization initialization schemes, such as non-clipped Gaussian initialization. However, this will require additional control on the asymptotics of $\|z_{in}\|$, $\|z_{in}\|_1$, and $V$. The result remains the same however.

## C.2 PROOF OF THEOREM 1

**Theorem 1** .[Feature Norm Growth in Linear Networks]
*Assume that the neural network is linear. Then, for any $\delta \in (0, 1/2)$, under the assumptions on the initialization stated in Theorem 2, there exists a universal constant $\lambda > 0$ such that for any $T$ and $\eta$ such that $T \leq \lambda \eta^{-1}$, the following holds with probability at least $1 - 2n^{-1+2\delta}$*

$$\sup_{1 \leq t \leq T} \left| n^{-1} \|W_t z_{in}\|^2 - \Gamma_t \right| \leq C n^{-\delta}, \tag{9}$$

*where $\Gamma_t = \Gamma_0 + \beta^2(1 + t(t-1))$, $\beta = \eta\, n^{-1} \|z_{in}\|_1$, and $\Gamma_0 = n^{-1} \|W_0 z_{in}\|^2$. In other words, $n^{-1}\|W_t z_{in}\|^2$ exhibits quasi-quadratic growth at early training phase, when the width is sufficiently large.*

*Proof.* Recall the update with SignSGD

$$W_{t+1} = W_t - \eta\, n^{-1} \chi_t\, \mathcal{S}(V) \otimes z_{in},$$

where $\mathcal{S}(.) = sign(.)$ and $\chi_t = \mathcal{S}(V^\top W_t z_{in} - \hat{y})$.

Denoting $\alpha_t = \langle W_t z_{in}, \mathcal{S}(V) \rangle$, we obtain

$$\alpha_{t+1} = \alpha_t - \beta \chi_t \times n = \alpha_0 - \beta n \sum_{j=0}^{t} \chi_t.$$

Therefore,

$$\begin{aligned}
\|W_{t+1} z_{in}\|_2^2 &= \|W_t z_{in}\|_2^2 + \eta^2 n^{-2} \|z_{in}\|_1^2 \times n - 2\eta n^{-1} \|z_{in}\|_1 \chi_t \times \alpha_t \\
&= \|W_t z_{in}\|_2^2 + \beta^2 \times n - 2\beta \chi_t \times \alpha_t.
\end{aligned}$$

Let $\delta \in (0, 1/2)$. From Theorem 2, it is straightforward that there exists a constant $\lambda > 0$ such that for any $T > 1$ and $\eta$ such that $T \leq \lambda \eta^{-1}$, with probability at least $1 - n^{-1+\delta}$, we have for all $t \leq T, \chi_t = \chi_0$. In this case, for $t \leq T$, we have $\chi_t \times \alpha_t = \chi_t \times \alpha_0 - \beta n \sum_{j=0}^{t-1} \chi_t \times \chi_j = \chi_t \times \alpha_0 - \beta n \times t$.

Therefore,

$$n^{-1} \|W_{t+1} z_{in}\|^2 = n^{-1} \|W_t z_{in}\|^2 + \beta^2 + 2\beta^2 t - 2\beta \chi_t n^{-1} \alpha_0.$$

Using Chebyshev's inequality, we can easily show that for any $\delta \in (0, 1)$, with probability at least $1 - n^{-1+2\delta}$, we have

$$|\chi_t n^{-1} \alpha_0| \leq \bar{Z} n^{-\delta},$$

which yields that with at least the same probability we have

$$\left| n^{-1} \|W_t z_{in}\|^2 - \Gamma_t \right| \leq \left| n^{-1} \|W_{t-1} z_{in}\|^2 - \Gamma_{t-1} \right| + 2\beta \bar{Z} n^{-\delta},$$

where we define the sequence $\Gamma_{t+1} = \Gamma_t + \beta^2(1 + 2t)$, with $\Gamma_0 = n^{-1} \|W_0 z_{in}\|^2$. Then, it is straightforward that for all $t \leq T$

$$\left| n^{-1} \|W_t z_{in}\|^2 - \Gamma_t \right| \leq 2\beta \bar{Z} T n^{-\delta}.$$

With union bound, this occurs with probability at least $1 - 2n^{-1+\delta}$. $\qquad\square$

Note that the probability bound can be significantly improved by considering sub-gaussian concentration bounds instead of Chebyshev's inequality. Since our aim in this paper is mainly methodological, we do not include it here.

# D ADDITIONAL EXPERIMENTAL DETAILS

## D.1 EXPERIMENTAL SETUP FOR THE LINEAR NETWORK

The linear network is given by

$$f(x) = W_2 W_1 W_0 x,$$

where $x \in \mathbb{R}^d$, $W_0 \in \mathbb{R}^{n \times d}$, $W_1 \in \mathbb{R}^{n \times n}$, and $W_2 \in \mathbb{R}^{1,n}$.

**Dimensions.** We use $d = n = 100$ in our experiments.

**Training Data.** We generate a random vector $w \in \mathbb{R}^d$ with iid coordinates $w_i \sim d^{-1/2} \mathcal{N}(0, 1)$ and fix it for the next step. Then, we generate $N = 1000$ samples from the following distribution:

- $x \sim \mathbb{R}^d$ random vector with iid coordinates $x_i \sim \mathcal{N}(0, 1)$
- $y = w^\top x + \epsilon$, where $\epsilon \sim \mathcal{N}(0, 0.025)$

**Training.** We use Adam algorithm for training, and train the model for $T = 300$ steps with full batch.

## D.2 EXPERIMENTAL SETUP FOR SFT (CLASSIFICATION)

For ANLI experiments, we use the following training configuration

- Training datasets: ANLI
- Training algorithm: AdamW, no warmup, linear schedule, dropout (0.1).
- Max sequence length 256.
- LoRA $\alpha = 2r$
- Precision: bf16.

We use $r = 8$ for MLP placement stratgy, and adapt $r$ to match param count for other placement strategies. Specifically:

- Qwen3.5-0.5B: MLP ($r = 8$), Attn ($r = 36$), PLoP($r = 17$)
- Llama3.2-1B: MLP ($r = 8$), Attn ($r = 27$), PLoP($r = 15$)

## D.3 EXPERIMENTAL SETUP FOR SFT (TEXT GENERATION)

**MetaMathQA.** For the SFT experiments on MetaMathQA we use the following training configuration

- Training dataset: MetaMathQA
- Training algorithm: Adam
  - epochs: 2
  - warmup: 0.1 fraction
  - schedule: cosine
  - no dropout
  - For each adapter placement we sweep over the learning rate in $\{1, 2, 3, 4, 5\} \times 10^{-4}$.
- Max sequence length 1024.
- LoRA $\alpha = 2r$
- Precision: bf16.

For evaluation on GSM8k we use the script `evaluate_chat_gsm8k.py` in the official QwenLM repo. We evaluate with 8-shot examples using the Qwen chat template. We apply a strict match for evaluating the accuracy and allow 512 generation tokens.

**SFT on Languages.**   For the SFT experiments on multilingual language modeling (AYA dataset) we use the following training configuration:

- Training dataset: Aya multilingual dataset (Portuguese, Japanese, Arabic)
- Models: Llama-3.2-1B-Instruct and Llama-3.2-3B-Instruct
- Training samples: 4,500 per language
- Training algorithm: AdamW
    - epochs: 50
    - warmup: 0.1 fraction (10%)
    - schedule: linear decay
    - learning rate: sweep over $\{1, 2, 3, 4, 5\} \times 10^{-5}$
    - weight decay: 0.01
- Batch size: 128
- Max sequence length: 128
- LoRA configuration:
    - rank $r = 16$ for MLP (adjusted for other methods to match param count)
    - $\alpha = 2r = 32$
    - No dropout
- Precision: fp16
- Evaluation metric: Negative log-likelihood (NLL) on held-out test set

**CommonsenseQA Experiments.**   For the SFT experiments on CommonsenseQA we use the following training configuration:

- Training dataset: CommonsenseQA
- Models: Qwen2.5-7B-Instruct, Qwen2.5-32B-Instruct
- Training samples: 9,500
- Training algorithm: AdamW
    - epochs: 50
    - warmup: 0.1 fraction (5%)
    - schedule: linear decay
    - learning rate: sweep over $\{1, 2, 3, 4, 5\} \times 10^{-5}$
    - weight decay: 0.01
- Batch size: 32 with gradient accumulation steps of 8 (effective batch size: 64)
- Max sequence length: 256
- LoRA configuration:
    - rank $r = 8$ for MLP (adjusted for other methods to match param count)
    - $\alpha = 2r = 16$
    - dropout: 0.05
- Precision: fp16
- Evaluation metrics: Negative log-likelihood (NLL) and multiple-choice accuracy on held-out test set (500 samples)

## D.4   GRPO EXPERIMENTS

For the GRPO experiments on MetaMathQA we use the following training configuration:

- Training dataset: MetaMathQA (50,000 samples)
- Training algorithm: AdamW

- – learning rate: $4 \times 10^{-6}$ (fixed, not swept due to computational constraints)
- – warmup: 0.1 fraction (10%)
- – schedule: cosine decay
- – weight decay: 0.01

- Batch size: 64 (16 per device with 4 gradient accumulation steps)

- Number of generations per prompt: 8

- Maximum generation length: 512

- Precision: bf16

- Reward functions: Combination of correctness and format rewards

- Evaluation: Custom GSM8K evaluation script using model-specific chat templates

**Note:** Unlike the SFT experiments, we could not perform learning rate sweeps for GRPO due to limited computational resources and the high cost of GRPO training runs. We expect that learning rate tuning could further improve the results.

## D.5 ADDITIONAL EMPIRICAL RESULTS

## D.6 SENSITIVITY ANALYSIS W.R.T SEQUENCE LENGTH

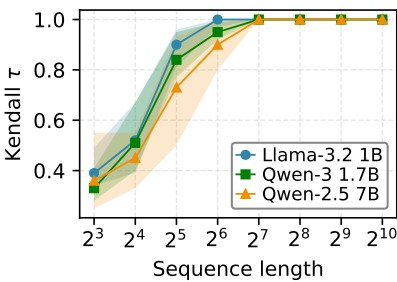

Figure 10: Sensitivity of the PLoP-picked module types as a function of sequence length.

**Sensitivity of** PLoP **to selection sequence length.**    To understand the impact of sequence length on PLoP-selected module types, we run the following experiment: we randomly select 2 batches of size 128 from TriviaQA (Joshi et al., 2017), trim the sequence length to a given value and compute the Kendall $\tau$ correlation coefficient between the module types selected by PLoP for each pair. We do this for 200 pairs of batches, and show the distribution of Kendall $\tau$ in Fig. 10 as a function of sequence length. Across different model sizes, we observe that correlation is close to 1 for sequene length $m \geq 128$, indicating that PLoP can be used relatively small sequence length.

## D.7 EVOLUTION OF NFN SCORE DURING LoRA FINETUNING

In Fig. 11, we show the evolution of the NFN scores as we finetuning a Llama3.2-1B model on Portuguese language dataset (see experimental setup below) for the three PLoP-selected module type: Value, Out_proj, and Down_proj. There are roughly 1750 fientuning steps, but we limit the curve to 1000 steps for better visualization. NFN scores increase with finetuning, which is in agreement with our theoretical predictions in Section 2.

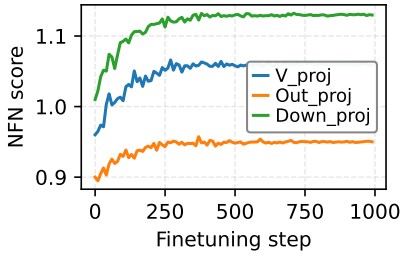

Figure 11: NFN evolution during finetuning.

## D.8 GRPO RESULTS FOR GEMMA3

Table 6: GRPO results for Gemma3-1B trained on MetamathQA (Yu et al., 2023).

| Module Types | Rwd/Format | Rwd/Answer | Eval/GSM8K |
|---|---|---|---|
| No RL | – | – | 29.10% |
| Attn (Q-K-V) ($r = 16$) | 2.16 | 0.89 | 30.05% |
| MLP (U-D-G) ($r = 16$) | 2.11 | 0.88 | 29.81% |
| PLoP$^{-1}$ (O-G-D) ($r = 16$) | 1.91 | 0.86 | 28.05% |
| PLoP(K-V-U) ($r = 16$) | 2.36 | 0.92 | **30.52%** |

Interestingly, for Gemma3 1B, we found that most of the RL rewards was accumulated in forms of format reward (placing the thinking process between `<think>` and `</think>` and the solution between `<answer>` and `</answer>`). This is reflected in Table 6. However, for eval on GSM8K, we found that accuracy after GRPO didn't change significantly which is probably due the fact that Gemma3-1B is weak on such tasks. In such cases, LoRA is probably not suitable, and full finetuning is needed to enhance reasoning capabilities.

## D.9 ADDITIONAL NFN-MAPS

### D.9.1 QWEN3-1.7B-INSTRUCT

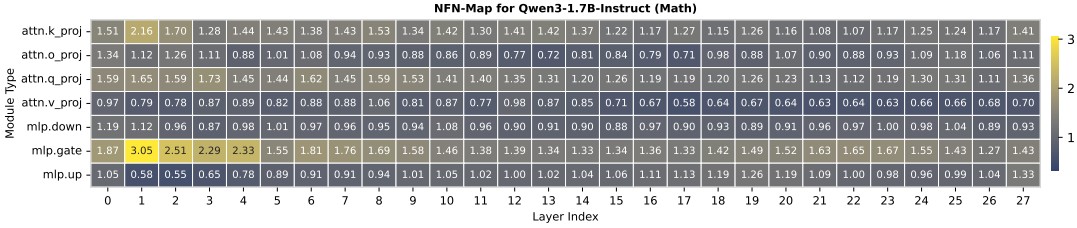

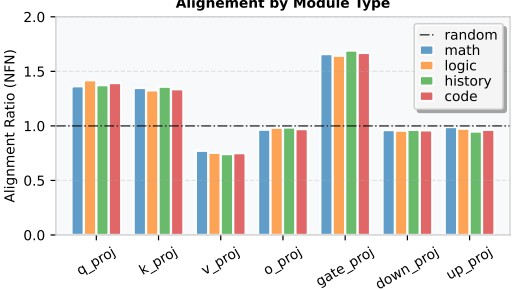

Figure 12: NFN scores for Qwen3-1.7B

## D.9.2  QWEN2.5-3B-INSTRUCT

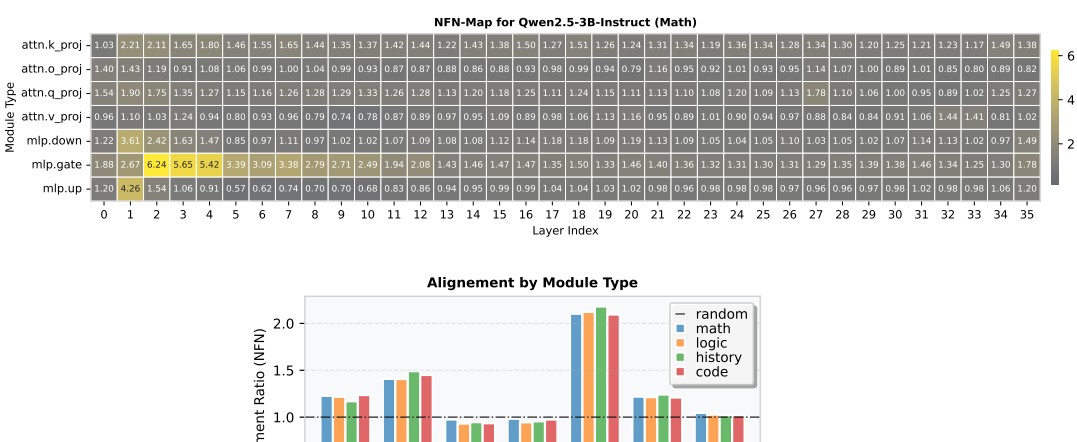

Figure 13: NFN scores for Qwen2.5-3B

## D.9.3  QWEN2.5-1.5B-INSTRUCT

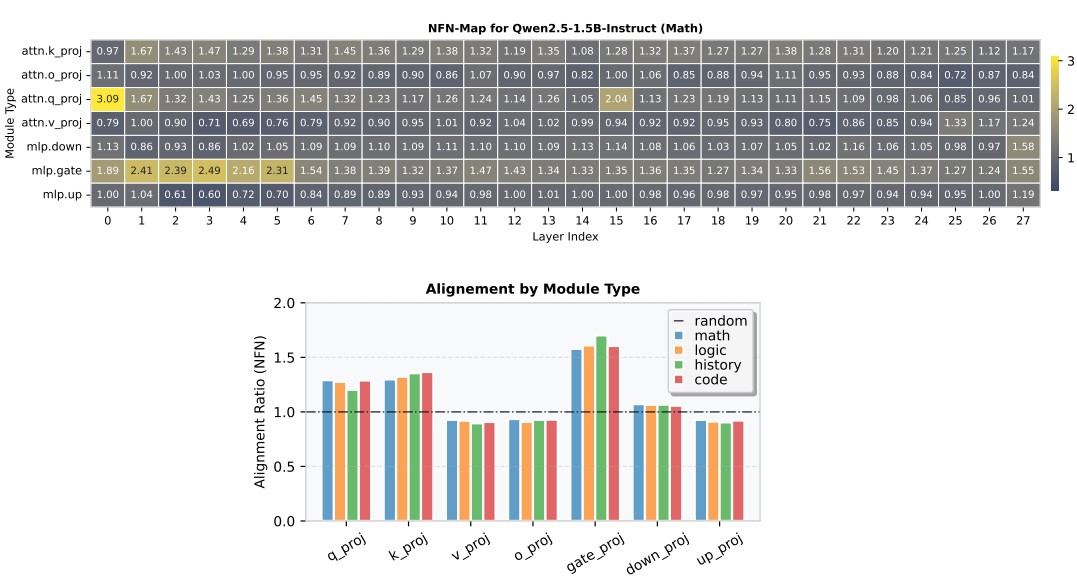

Figure 14: NFN scores for Qwen2.5-1.5B

### D.9.4 QWEN2.5-1.5B-CODER-INSTRUCT

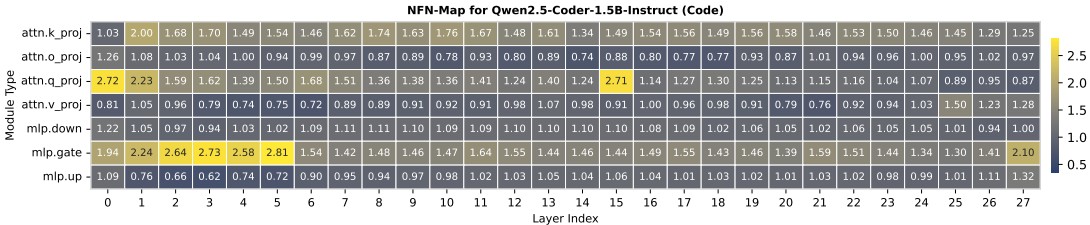

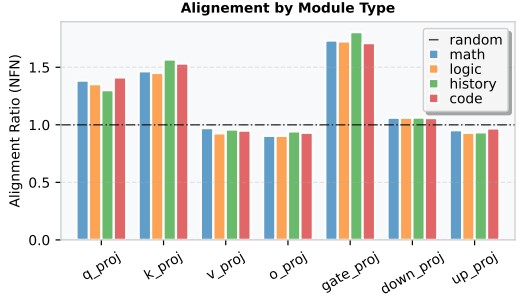

Figure 15: NFN scores for Qwen2.5-1.5B-Coder

### D.9.5 GEMMA3-1B-INSTRUCT

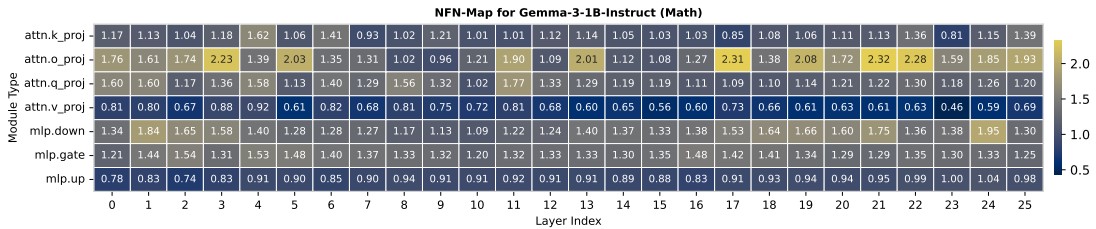

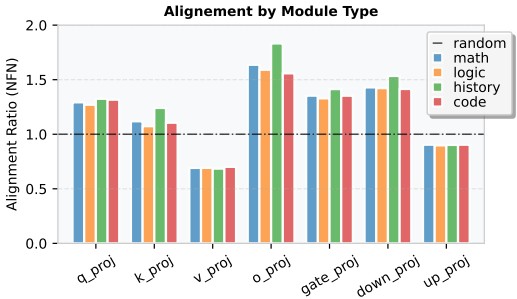

Figure 16: NFN scores for Gemma3-1B-Instruct

