# OpenReview forum: "PLoP: Precise LoRA Placement for Efficient Finetuning of Large Models"
_ICLR.cc/2026/Conference — ICLR 2026 Poster_

### Official Review · Reviewer_VYDu · 2025-10-21

**Soundness:** 3
**Presentation:** 3
**Contribution:** 3
**Rating:** 6
**Confidence:** 4

**Summary:**

This paper introduces Precise LoRA Placement, a lightweight method to determine where to place LoRA adapters. Specifically, this paper proposes a alignment metric based on feature norm that identifies the modules requiring fine-tuning by performing inference on a small number of samples from downstream datasets. Extensive experiments demonstrate the effectiveness of the proposed method.

**Strengths:**

1. The proposed metric appears to be lightweight and efficient.

2. The proposed method has demonstrated its effectiveness through extensive experiments.

3. The availability of code enhances the reproducibility of the work.

**Weaknesses:**

1. The experimental section only compares the proposed method with the original LoRA, without including other LoRA variants or approaches that also explore efficient LoRA adaptations.

2. The definition of the Normalized Feature Norm is confusing. The motivation for introducing a random variable here, rather than directly normalizing by the product of the norms of two vectors (or matrices), is not clearly explained.

3. The results in Figure 4 show that $v_proj$ consistently exhibits lower alignment score across different models, which is an interesting observation. Providing an explanation for this phenomenon could deepen the understanding of the proposed metric and enhance the overall depth of the paper.

4. Directly reporting the computation time and memory usage required for the proposed metric would provide a clearer and more intuitive demonstration of the method’s efficiency.

**Questions:**

1. According to the definition of the proposed metric, it is possible to directly compute the score for each module in every layer. Why not adopt a more fine-grained localization strategy, instead of grouping modules such as Q, K, and V together for unified localization?

2. The results presented in Figure 5 appear to contradict the intuition behind the proposed method. The paper suggests that modules with lower alignment are more suitable for fine-tuning, yet Figure 5 shows that, for task-specific models, the largest score improvements (compared to general model) occur in the modules that were initially the highest, rather than in the low-alignment ones.

3. It would be helpful to show the dynamic changes of the proposed metric during the fine-tuning process.

---

> ### Author Response · Authors · 2025-11-20
> **Authors Response [1/2]**
>
> We thank the reviewer for their feedback and for the positive assessment of our work. Below we address each comment and question, and indicate how the revised version has been updated.
>
>
> ## Comments
> ---
> ### 1. Other LoRA variants
> The reviewer mentioned that *"The experimental section only compares the proposed method with the original LoRA, without including other LoRA variants or approaches that also explore efficient LoRA adaptations."*
> We believe there is a misunderstanding on this point. In this work, we introduce PLoP, an adapter placement strategy for LoRA finetuning whose goal is to *automatically select module types for LoRA*. Thus, we only compare with other common *placement strategies* for LoRA such as MLP and Attn. The gains from PLoP are largely orthogonal to gains from other variants of LoRA (e.g. DoRA, LoRA+, PiSSA, etc) and one should expect these improvements to compound. However, we consider this beyond the scope of this paper.
>
>
> ### 2. Random baseline in NFN score
> The reviewer suggested that "*The definition of the Normalized Feature Norm is confusing*". Following the reviewer’s suggestion, we reorganized Section 3 so that we first motivate and introduce the random baseline feature norm, explaining why a “random” vector $\tilde{z}_{in}$ should yield NFN scores around 1, and why this serves as a meaningful “no-alignment” reference point. Only afterwards do we present the NFN definition.
> Regarding the reason we chose the random baseline instead of direct normalization: both versions are cheap to compute, but we prefer the current one because it defines NFN as a ratio to a random-alignment baseline. This is more an aesthetic choice than anything else, and both expressions yield the same results. We added a short paragraph describing this in Section 3.
>
> ### 3. Low NFN scores for v_proj
> Thanks for raising this interesting observation. We have indeed found that the Value, Out_proj (excluding Gemma), and Up_proj modules exhibit low NFN scores accross different models and datasets, as compared to e.g. Query/Key modules. We currently do not have a satisfying explanation for why this happens, but it is likely related to details of pretraining and the data mix. Without such information, it is difficult to make reliable hypotheses.
>
>
> ### 4. Compute cost of PLoP
> Regarding compute cost of PLoP, we have made this explicit in our paper (see Compute cost paragraph in page 7).
> - Compute cost: PLoP requires a single forward pass with a modest batch size (200 in our experiments), no gradients, and no extra model variants. This is comparable to a standard batched prefill and negligible relative to full finetuning or even LoRA training itself.
> - *Stability vs. subset size*: We added a sensitivity analysis (Fig. 6), showing that the selected module types stabilize with batches of size $\approx$64 samples, which further reduces the required computation.
> - *Stability vs. sequence length*: in Fig D.6 in the appendix, we show that PLoP can be effective with sequence lengths as low as 128.
>
> Thus, PLoP can be used with large models while preserving the core benefit of LoRA: saving compute and memory.
>
> ## Questions
> ---
>
> ### 1. Understanding the result in Fig 5
>
> Thank you for raising this. Our method uses NFN scores to decide where to place LoRA: when we apply PLoP, we evaluate the model on the task dataset and pick module types with the lowest NFN, because they are the least aligned with the new task and thus have the most “unused capacity”. In contrast, Figure 5 is showing something different: It compares NFN of a general Qwen2.5 model to NFN of a fully task-specialized model (Qwen2.5-Math / Qwen2.5-Coder), and looks at the change in NFN per module type after a *long task-specific full-finetuning run in which all weights are updated*. So PLoP is based on where NFN is currently low, while Figure 5 is about where NFN happens to increase the most during a long, unconstrained training trajectory (Full parameter training). These are related but not the same, and our theory never claims that “the largest NFN increase must occur in the lowest-alignment modules.”
>
>
> ### 2. Changes of NFN during fine-tuning
> We thank the reviewer for this interesting suggestion. We included a **new experiment** in the revised paper showing how NFN scores change as we finetune a Llama3.2-1B model with LoRA on a language dataset. Please see **section D.7 and Fig 11** in the appendix for more details. The results show an increase in NFN scores as we finetune the model, as expected from our theoretical analysis (Sec 2).

---

> > ### Author Response · Authors · 2025-11-20
> > **Authors Response [2/2]**
> >
> > ### 3. Why not adopt a more fine-grained approach?
> > The reviewer raised an important question. We do not consider module-level selection for two reasons:
> > - In Fig 3, we show an example of the most fine-grained NFN scoring, which could be used to perform per-module selection. However, selecting individual modules (instead of module types) with the lowest NFN scores leads to irregular patterns. For instance, in some layers, very few or none of the modules are selected, while in other layers, most modules are selected. This tends to hurt performance, and we found that module-type level selection is more robust to extremes.
> > - In practice, LoRA finetuning is usually conducted with per-type selection rather than fine-grained selection, and PLoP provides an automatic module-type selection method.
> >
> >
> > ## Additional Experiments
> > ---
> > In the revised paper, we added new experiments and improved existing ones so that PLoP’s benefits are more visible:
> > - *Math SFT (MetaMathQA → GSM8K)*:
> >         - Qwen3-0.6B: PLoP achieves 63.8%, beating the second best performance 63.3% with MLP, at matched parameter count.
> >         - Qwen3-1.7B: PLoP reaches 75.4%, better than MLP (75.0%) at the same param count, and clearly better than ALL (73.9%) at much smaller param count.
> > - (**New Experiments**) **Language SFT (AYA languages, Llama-3.2 1B & 3B)**: PLoP consistently achieves the lowest test NLL across Portuguese, Japanese, and Arabic, with the second-best method almost always being MLP. The Attn and Random baselines lag behind, and PLoP$^{-1}$ (selecting highest NFN scores) performs significantly worse, confirming that NFN is capturing meaningful signal rather than noise.
> > - (**New Experiments**) **CommonsenseQA (Qwen-2.5 7B & 32B)**: PLoP improves test accuracy over all alternatives on both 7B and 32B models (e.g. 92.6% vs 91.95% for the second best baseline on Qwen-2.5 32B).
> > - *Classification on ANLI (Fig. 7)*: For Qwen2.5-0.5B and Qwen3-Embedding-8B, PLoP outperforms ATTN and MLP; for Llama-3.2-1B, PLoP is competitive with MLP and significantly better than Attn, consistent with NFN suggesting that both PLoP and MLP target low-alignment modules.
> > - *RL with GRPO on MetaMathQA (Qwen3-1.7B)*: PLoP yields higher reward and GSM8K accuracy than both Attn and MLP, even when using fewer trainable parameters (e.g., with rank 16, 6.88M PLoP params vs 11.01M MLP params).
> > Across these settings, PLoP yields consistenly better performance compared to other placement heuristics at negligible cost.
> > - *NFN-based sanity check*: we use PLoP$^{-1}$ (choosing module types with the highest NFN scores) as a sanity check baseline. In all experiments, PLoP$^{-1}$ performs noticeably worse than PLoP and other heuristics, confirming that module-data alignment measured by NFN is informative for adapter placement and that PLoP’s gains are not due to random variance.
> >
> > These results indicate that module choice does matter, and that PLoP provides a consistent, low-cost way to choose modules, that is consistently better than other heuristics.
> >
> >
> > We hope our response and revision addresses the reviewer's main concerns.

---

> > > ### Comment · Reviewer_VYDu · 2025-11-26
> > >
> > > Thank you for your response. Your reply has addressed most of my concerns!
> > >
> > > Regarding the first question, I hope the paper can experimentally demonstrate that the proposed method is effective across different LoRA variants, as this would better substantiate its robustness.
> > >
> > > In addition to LoRA variants, I would also like to see a comparison between the proposed method and other position selection strategies, such as [1][2].
> > >
> > > [1] Interpreting and Improving Large Language Models in Arithmetic Calculation, ICML 2024.
> > >
> > > [2] HeadMap: Locating and Enhancing Knowledge Circuits in LLMs, ICLR 2025

---

> > > > ### Author Response · Authors · 2025-11-28
> > > > **Authors response**
> > > >
> > > > We thank the reviewer for their feedback and we are glad our rebuttal addressed most of the concerns.
> > > >
> > > > While we currently believe that experimenting with LoRA variants is out of the scope of this paper, we will definitely explore this direction in future work. Regarding the proposed methods mentioned by the reviewer, *they are not module-type level selection methods*: [1] and [2] are based on circuits, and their methodology identifies relevant attention heads for the task, while PLoP identifies which module types (Query, Key, Value, Up_proj, Down_proj, etc) are best for LoRA finetuning. We compare with commonly used module-type level selection strategies as detailed in the paper. However, we believe they're somewhat related and we will include them in the related work section.

---

### Official Review · Reviewer_Goqy · 2025-10-23

**Soundness:** 3
**Presentation:** 2
**Contribution:** 2
**Rating:** 6
**Confidence:** 4

**Summary:**

The article under review proposes PLOP, an inexpensive, task-aware way to place LoRA adapters by ranking module types with a simple alignment score and inserting LoRA where alignment is lowest.  The authors use a normalized feature norm (nfn) that compares a module’s feature magnitude on real inputs to a randomized-input baseline:
$$
\mathrm{NFN}(W,x)=\frac{\lVert W z_{\text{in}}(x)\rVert}{\lVert W \tilde z_{\text{in}}(x)\rVert}
\approx
\frac{\lVert W z_{\text{in}}(x)\rVert}{\lVert W\rVert_F,\lVert z_{\text{in}}(x)\rVert}
$$
$(\tilde z_{\text{in}})$ shares the norm of $(z_{\text{in}})$ but has i.i.d. gaussian coordinates. higher nfn means stronger module–data alignment; scores near 1 mean weak alignment. plop averages nfn per module type and places LoRA adapters in the lowest-scoring types. Some theoretical justifications behind the method are also provided.

**Strengths:**

1. The paper reframes adapter placement as a task-aware alignment problem and proposes a simple, gradient-free proxy (normalized feature norm) to rank module types. this is a clean definition that removes scale effects via a randomized baseline, making it broadly applicable across architectures and tasks. the theory–metric–procedure chain is coherent and distinct from gradient/sensitivity-based selection, which is heavier and less LoRA-friendly.

2. Theoretical pieces motivate why feature norms should grow with alignment and width, then connect that to a practical nfn estimator; even if idealized, the analysis yields testable predictions that the experiments largely reflect. the empirical study spans sft classification, sft math generation, and rl (grpo) with careful parameter-matching, showing consistent wins or parity against attention-only and mlp-only baselines, sometimes beating “all modules” at lower budget. compute claims are credible and supported by explicit cost discussions and a one-batch nfn recipe.

3. The mechanism is easy to follow: compute nfn per type, pick the lowest, insert lora. the paper repeatedly reminds the reader what the type codes mean (q, k, v, o, u, g, d), provides nfns as maps and per-type bar charts, and summarizes training configs in appendices. figures and tables tie back to claims (e.g., llama3.2 parity with mlp explained by low mlp nfn; qwen math gains with d–o–v). overall presentation is minimal and readable.

4. Adapter placement is a common pain point with real budget constraints; a method that costs roughly one forward pass and often matches or improves over prevailing heuristics is practically valuable.

**Weaknesses:**

1. The finetuning tasks are confined to anli and math (metamathqa to gsm8k) for sft and grpo; there are no non-math generative, instruction-following, multilingual, or long-context evaluations. while fig. 4–5 report nfn patterns for code/history/logic, there are no finetuning experiments on those domains, so generality remains untested.

2. NFN is computed from a single forward pass using hooks, but the paper does not report estimator variance, sensitivity to batch/sequence length, or across-batch stability of type rankings. without confidence intervals or stability curves, it is hard to know how often plop would pick the same types on a new sample.

3. The authors note that layer-level placement using the nfn-map gave “inconsistent results” and defer it, yet per-layer or per-head selection is exactly where a cheap ranking could shine; not probing this systematically is a weakness.

**Questions:**

1. How stable are nfn rankings in practice? Please report bootstrap confidence intervals and a stability curve (kendall-τ vs number of tokens/batches) for per-type nfn, plus any sensitivity to seq length.

2. Can you give a targeted layer-level ablation using the nfn-map? Choose the bottom-k individual layers by nfn and compare against type-level plop under the same parameter budget; report when layer-level helps or hurts, and whether an nfn-gap threshold or per-type cap stabilizes results. this would clarify the “inconsistent results” remark and indicate how to operationalize finer placement.

---

> ### Author Response · Authors · 2025-11-20
> **Authors Response**
>
> We thank the reviewer for theIR feedback, and for the positive assessment of our work. Below we address each question, and we indicate how the revised version has been updated.
>
> ### 1. Finetuning tasks and models
>
> While in our original experiments we chose to focus on ANLI and mathematical reasoning (MetaMathQA and GSM8K) for both SFT and RL, in the revision, we have broadened the experimental coverage beyond ANLI and MetaMathQA, and included larger scale models (Qwen2.5-32B):
> - **SFT for language modeling (Table 3)**: We now finetune Llama-3.2 1B and 3B models on three AYA language datasets (Portuguese, Japanese, Arabic) and report train/test NLL. PLoP consistently achieves the lowest test NLL at matched parameter budgets, outperforming Attn, MLP, and Random selection.
> - **Commonsense QA (Table 4)**: We added experiments on CommonsenseQA with Qwen-2.5 7B and 32B, evaluating answer accuracy. Here again, PLoP improves over attention-only, MLP-only, and random selection baselines, while having the same adapter parameter budget. This also demonstrates that PLoP scales to large (32B) models, beyond the small/medium regime.
> - Reinforcement learning: In addition to GRPO on MetaMathQA → GSM8K with Qwen3-1.7B, we include extra GRPO runs (Gemma3-1B) in the appendix. Across these RL experiments, PLoP achieves higher reward and GSM8K accuracy than Attn and MLP.
>
> These additional experiments provide a broader experimental setup and the show the consistent benefits of using PLoP. Due to compute constraints, we could not run any additional experiments.
>
>
> ### 2. Stability of NFN estimates and rankings
> The reviewer commented that “*NFN is computed from a single forward pass… no estimator variance, sensitivity to batch/sequence length, or across-batch stability of type rankings. Please report bootstrap CIs and a stability curve (Kendall-τ vs number of tokens/batches).*”
>
> We agree that understanding how many samples and tokens are needed for a reliable NFN ranking is important for practitioners. In our experiments, we found that a sequence length of 256 and sample size of 200 yield stable selections.
>
> **(a) Why we expect stability**: By construction, per-type NFN is an average over many activations:
> - Over tokens in the dataset subset,
> - Over positions within each sequence,
> - And over all modules of that type.
>
> This heavy averaging suppresses variance, and PLoP uses only the ordering of types (argmin over NFN), not the raw magnitudes. In tasks where the NFN gaps between types are large, this makes the ranking robust to sampling noise.
>
> **(b) What we have added**:  In the revised paper, we clarify that NFN is computed on a single, moderate-size subset (batch size ≈ 200, small sequence lengths of 256) and that this single pass is sufficient in all our experiments to obtain consistent PLoP choices across runs.
>
> Following the reviewer’s suggestion, we added two new stability/sensitivity experiments: Sensitivity of PLoP-selected module types to the size of selection subset (Fig 6) and the sequence length (Sec D.6 in the appendix). These plots show the distribution of Kendall $\tau$ correlation coefficient.
>
>
> ### 3. Layer-level placement
>
> We appreciate this suggestion and agree that layer-level selection could be a natural extension of our framework. As we mentioned in the paper, we conducted some experiments with layer-level placement and found inconclusive results. Here is why we think layer-level selection didn't work: Looking at NFN-map (the most granular level of NFN scoring, see e.g. Fig 3 in the paper), we can see that in each layer, there are low-NFN score module types with similar NFN scores accross layers. By performing a layer-level selection will necessarily discard some of these "important" modules leading to inconsistent performance. We are still evaluating this phenomenon which we leave for future work.
>
>
>
> We are grateful for the reviewer’s careful reading and interesting suggestions (stability analysis and finer-grained placement) that can make NFN-based PLoP even more useful in practice. We hope the revised paper addresses the reviewer's main concerns.
>
>
> -------

---

### Official Review · Reviewer_VHUP · 2025-10-30

**Soundness:** 3
**Presentation:** 1
**Contribution:** 2
**Rating:** 4
**Confidence:** 3

**Summary:**

The paper proposes a method called PLoP for dynamically selecting LoRA training modules through precomputed metrics. Specifically, PLoP first defines an indicator, namely NFN, to measure the alignment degree between model modules and training data, and selects the modules for inserting LoRA adapters based on this indicator. The NFN score of each module is obtained by dividing the output from the module with the original data features by the output from the module with random features.

**Strengths:**

1) The proposed method exhibits strong generalization capabilities. It presents a unified training framework that can be generalized to any large model training approach. By simply providing the model and dataset, computing the NFN metrics, and then freezing modules based on these metrics, the method can be applied universally.

2) The theoretical analysis is comprehensive. The design motivation of NFN is supported by the theoretical basis of "Feature Norm Growth in Linear Networks", making it theoretically reasonable.

**Weaknesses:**

1) The most unacceptable issue with this work is that the experimental results are not significant. In the main experiments, there is no substantial difference between PLOP and fine-tuning other modules, which may indicate that researching which module is more worthy of fine-tuning has little value.

2) As a method for saving computational resources, what I expect from PLoP is its efficiency in the efficient fine-tuning of large-parameter models. However, the experimental part only uses small models below 7B.
3) The datasets used are relatively single. Consistent with the first point, since NFN calculation depends on specific datasets and models, the generalization ability of PLoP may lack verification on multiple datasets.
4) It is recommended to optimize the typesetting of Figure 8's position.
5) Figure 2 and Figure 8 are duplicated. Should it be corrected to Figure 2 in Line 191 ?

**Questions:**

1) PLOP is equivalent to fine-tuning one of the modules via LoRA. Why didn't the authors indicate which module corresponds to PLOP in the experiments?

---

> ### Author Response · Authors · 2025-11-20
> **Authors Response [1/2]**
>
> We thank Reviewer VHUP for the careful reading and constructive feedback. Below we address each concern in turn and summarize the changes made in the revised manuscript (changes are highlighted in blue in the uploaded version).
>
> ### 1. “The experimental results are not significant… maybe module choice has little value.”
>
> We respectfully disagree and have strengthened the empirical section to make the benefits of PLoP clearer.
> - **a) Clarifying the goal of PLoP.**
> The goal of PLoP is to provide a **principled**, automatic module-type selection rules that matches or **improves upon the strongest heuristic baselines (ATTN, MLP)**, works across a wide range of models and tasks, and has negligible overhead. Specifically, PLoP can be merged with the first forward pass in LoRA finetuning, which basically implies that PLoP selects module type at **virtually zero cost**.
> Even if PLoP is only “as good as the best hand-tuned choice” in some settings, this is already valuable: **it eliminates ad-hoc trial-and-error** over module placements, which is expensive for large models.
>
> - **b) Improvements in the experiments section in the revised paper.**
> We could not run MetaMathQA finetuning with 32B model because of limited GPU resources. However, following the reviewer's request, we conducted some experiments with a 32B model on a smaller dataset (CommonsenseQA) that showed consistent gains with PLoP (see CSQA experiments below, and in the revised paper). In the revised paper, we added new experiments and improved existing ones so that PLoP’s benefits are more visible:
>     - Math SFT (MetaMathQA → GSM8K):
>         - Qwen3-0.6B: PLoP achieves 63.8%, beating the second best performance 63.3% with MLP, at matched parameter count.
>         - Qwen3-1.7B: PLoP reaches 75.4%, better than MLP (75.0%) at the same param count, and clearly better than ALL (73.9%) at much smaller param count.
>     - (**New Experiments**) **Language SFT (AYA languages, Llama-3.2 1B & 3B)**: PLoP consistently achieves the lowest test NLL across Portuguese, Japanese, and Arabic, with the second-best method almost always being MLP. The Attn and Random baselines lag behind, and PLoP$^{-1}$ (selecting highest NFN scores) performs significantly worse, confirming that NFN is capturing meaningful signal rather than noise.
>     - (**New Experiments**) **CommonsenseQA (Qwen-2.5 7B & 32B)**: PLoP improves test accuracy over all alternatives on both 7B and 32B models (e.g. 92.6% vs 91.95% for the second best baseline on Qwen-2.5 32B).
>     - Classification on ANLI (Fig. 7): For Qwen2.5-0.5B and Qwen3-Embedding-8B, PLoP outperforms ATTN and MLP; for Llama-3.2-1B, PLoP is competitive with MLP and significantly better than Attn, consistent with NFN suggesting that both PLoP and MLP target low-alignment modules.
>     - RL with GRPO on MetaMathQA (Qwen3-1.7B): PLoP yields higher reward and GSM8K accuracy than both Attn and MLP, even when using fewer trainable parameters (e.g., with rank 16, 6.88M PLoP params vs 11.01M MLP params).
> Across these settings, PLoP yields consistenly better performance compared to other placement heuristics at negligible cost.
>     - NFN-based sanity check: we use PLoP$^{-1}$ (choosing module types with the highest NFN scores) as a sanity check baseline. In all experiments, PLoP$^{-1}$ performs noticeably worse than PLoP and other heuristics, confirming that module-data alignment measured by NFN is informative for adapter placement and that PLoP’s gains are not due to random variance.
> Taken together, these results indicate that module choice does matter, and that PLoP provides a consistent, low-cost way to choose modules that is consistently better than other heuristics.
>
>
> ### 2. “Only small models (<7B) are used; I expected tests on larger models.”
>
> We fully agree that scalability to larger models is important. We believe that models of size 8B are already large enough. However, in the revised paper, we added finetuning experiments on **CommonsenseQA with Qwen-2.5 7B and Qwen-2.5 32B (Table 4 in the revised paper)**: PLoP achieves the best accuracy at a matched parameter budget, showing that our method scales to 32B-parameter models in realistic settings.

---

> ### Author Response · Authors · 2025-11-20
> **Authors Response [2/2]**
>
> ### 3. “Datasets are relatively single; generalization ability may lack verification on multiple datasets.”
> We agree that diversity of datasets is crucial to assess generalization. Our original paper included experiments with ANLI, MetaMathQA, and GSM8K. The revised paper now includes a broader and more varied suite of tasks and models:
> - SFT (Section 4.1–4.2): ANLI (challenging NLI) with Qwen2.5-0.5B and Llama-3.2-1B and Qwen3-Embedding-8B, MetaMathQA → GSM8K (math reasoning) with Qwen3-0.6B and Qwen3-1.7B, AYA language datasets (Portuguese, Japanese, Arabic) with Llama3.2-1B and Llama3.2-3B, and CommonsenseQA with Qwen2.5-7B and Qwen2.5-32B models.
> - Reinforcement learning (Section 4.3): GRPO on MetaMathQA with evaluation on GSM8K.
>
> Across all of these varying datasets and models, PLoP consistently outperforms or matches the best heuristic baseline. We believe this addresses the concern about generalization and demonstrates that PLoP is not tailored to a single dataset or domain.
>
>
> ### 4. Figure and typesetting issues (Fig. 8 position; duplication of Fig. 2 & Fig. 8; incorrect reference).
> We thank the reviewer for pointing out these issues. The duplication between Fig. 2 and Fig. 8 has been corrected.
> The reference “Figure 2 in Line 191” has been corrected to point to the intended figure.
>
>
> ### 5. “PLOP is equivalent to fine-tuning one of the modules… Why not indicate which module corresponds to PLoP?”
> We believe there is a misunderstanding on this point. First, PLoP selects module types, not a single module. PLoP operates at the level of module types (e.g., Query, Key, Value, Up, Down, Gate, Out), which are then instantiated across all layers. This is the standard granularity used in practice by LoRA libraries and practitioners, and is aligned with prior work (Hu et al., 2022; He et al., 2021).
>
> We also explicitly state the selected module types in each experiment. In all main tables, the PLoP configuration is clearly labeled, for example:
> - MetaMathQA → GSM8K, Qwen3-0.6B (Table 1): PLoP(D–U–V) indicates that PLoP selected Down_proj, Up_proj, and Value module types.
> - MetaMathQA → GSM8K, Qwen3-1.7B (Table 2): PLoP(D–O–V) indicates Down_proj, Out_proj, and Value.
> - RL (Table 5) similarly specifies the PLoP-selected types (V-O-D).
> This makes it explicit which module types PLoP chooses in each setting.
>
> Why these module types?
> The chosen module types are precisely those with lowest NFN scores, as described in the PLoP procedure in Section 3:
> Step 1: compute NFN scores per module type,
> Step 2: insert LoRA in types with lowest NFN.
> The NFN maps (Fig. 3–5) illustrate these patterns visually for various model–task pairs.
>
> We have also clarified in the text that PLoP$^{-1}$ is defined by choosing the highest-alignment module types, explicitly listing which types those are in each experiment.
>
> ### 6. Clarifying compute/efficiency expectations
> Regarding compute cost of PLoP, we have made this explicit in our paper (see Compute cost paragraph in page 7).
> - Compute cost: PLoP requires a single forward pass with a modest batch size (200 in our experiments), no gradients, and no extra model variants. This is comparable to a standard batched prefill and negligible relative to full finetuning or even LoRA training itself.
> - Stability vs. subset size: We added a sensitivity analysis (Fig. 6), showing that the selected module types stabilize with batches of size $\approx$64 samples, which further reduces the required computation.
> - Stability vs. sequence length: in Fig D.6 in the appendix, we show that PLoP can be effective with sequence lengths as low as 128.
> Thus, PLoP can be used with large models while preserving the core benefit of LoRA: saving compute and memory.
>
>
> The revised paper strengthens and clarifies the empirical significance of PLoP through additional experiments (including 7B and 32B models and multiple new datasets), makes module-type choices under PLoP completely explicit in all tables, fixes the figure/formatting issues highlighted by the reviewer, and clarifies both the theoretical motivation and the computational advantages of NFN-based placement.
> We appreciate the reviewer’s comments, which helped us improve the paper, and we hope that the revisions address the concerns raised.

---

> > ### Comment · Reviewer_VHUP · 2025-11-25
> >
> > Thanks for the authors' detailed response. I appreciate the authors' efforts in the rebuttal and the additional experimental results provided. I acknowledge that the problems solved by the authors are more valuable than I originally thought, and the supplementary experiments conducted by the authors have addressed my concerns about the generalization ability of PLoP. Overall, many of my concerns have been addressed.

---

> ### Author Response · Authors · 2025-11-26
> **Thank you**
>
> We thank the reviewer for their feedback and for raising their score form 4 to 6. We are happy to address any additional concerns.

---

### Official Review · Reviewer_vSmF · 2025-11-06

**Soundness:** 3
**Presentation:** 2
**Contribution:** 3
**Rating:** 4
**Confidence:** 4

**Summary:**

This paper introduces PLoP, a method for dynamically assigning LoRA adapters to different module types given pretrained models and fine-tuning tasks based on the normalized feature norm (NFN) score. The authors provide theoretical and empirical justification for the NFN score as a measure of module-data alignment. PLoP is derived from the NFN score to identify modules that benefit most from SFT. Computing MFN is done via a few forward passes (without gradients etc). The authors provide results to prove the benefit of PLoP on LoRA fine tuning for classification, text generation, and mathematical reasoning using accuracy as a metric.

**Strengths:**

Authors outline the importance of adapter placement, and show a computationally tractable and architecture-agnostic way of doing so.

Theoretical justification of NFN scores is compelling.

Experiments shed light on the variance of MFN scores between layers, modules, and architectures (with and without reasoning), indicating that there is a need for a method that takes that variance into account.

The proposed framework and contributions are novel to my knowledge.

Parameter-efficient finetuning methods are very timely (especially for discussions of continual learning).

**Weaknesses:**

In Text:
- Figure 1 needs more clarification on inputs and outputs, baseline feature norm.
- In section 3, first motivate and introduce random baseline feature norm, then introduce NFN formulation to improve clarity.
- The second and third paragraph in the Introduction can be tied together and streamlined. They also cover almost exactly the same content as related works.
- Please elaborate more on the baseline presented in section 3/figure 4 and 5
- Authors need to improve formatting in Appendix.

Empirical support for PLoP as a framework is lacking:
- It is not fully clear to me why the specialized Qwen Math model has higher NFN scores on a Math dataset; more focus on this could strengthen empirical justification
- PLoP^{-1} serves as a weak baseline, I would expect to see current SOTA selection methods (if existing), random selection, other scoring heuristics, or oracle module placement as baselines (this could be found by running a grid search over the modules of only one layer (for simplicity) of a 1B model, and comparing to the best selection).
- The improvements offered by PLoP in SFT for text generation are not super clear in section 4 (the authors are ablating over multiple parameters in the same table, unclear which numbers to compare); it took me quite a bit of time to understand the benefit PLoP offers over the baseline; could use better highlighting or illustrations to emphasize improvements.
- Baselines that are lacking: full finetuning, base model performance, gradient-based selection methods.
- Currently, the experiments section is hard to follow. Details on experimental setup and motivation are intertwined with the results, which is distracting.
- Please provide more details on computational complexity of PLoP and how it compares to methods that take gradients.

**Questions:**

I'd like to better understand the rationale behind the chosen baselines.

---

> ### Author Response · Authors · 2025-11-20
> **Authors Response [1/2]**
>
> We thank the reviewer for their feedback. We have revised the paper in response to the reviewer's comments. Below we address each point, and all changes are highlighted in blue in the revised version of the paper which is now accessible on Openreview.
>
>
> ### I. Text-related comments
> ----
> 1. *Fig1*: We agree that the original Fig 1 and caption did not fully clarify the inputs/outputs and the role of the baseline feature norm. In the revised version, we updated Fig 1 to explicitly define $z_{in}$ and $\tilde{z}_{in}$, and illustrate how PLoP computes NFN scores and uses them to select module types. We also expanded the caption to explain the computation of the alignment score.
>
>
> 2. *NFN definition*: Following the reviewer’s suggestion, we reorganized Section 3 so that we first motivate and introduce the random baseline feature norm, explaining why a “random” vector $\tilde{z}_{in}$ should yield NFN scores around 1, and why this serves as a meaningful “no-alignment” reference point. Only afterwards do we present the NFN definition.
>
>
> 3. *Baselines in Sec 3 (Fig 4 and 5)*: We clarified in the main text and captions that:
>     * Fig 4 and Fig 5 show NFN scores by module type, aggregated over all modules W of a given type.
>     * Each boxplot corresponds to a pair (model, dataset). For datasets, we use GSM8K for math, HumanEval for code, MMLU-history for history, and MMLU-logical-fallacies for logic. For each module type, we compute NFN scores over the dataset and average over all modules of that type, leading to the scores shown in the boxplots.
>     * The random baseline NFN = 1 is plotted as the reference “no-alignment” line.
> We would like to refer the reviewer to the box describing PLoP at the bottom of page 5, where we explain how module-type NFN scores are obtained.
>
> 4. *Formatting issues in the Appendix*: we thank the reviewer for spotting this issue. We have now fixed the formatting issue in the appendix (see revised version).
>
>
>
>
> ## II. Empirical results
> ----
> ### 1. Why does Qwen-Math have higher alignement with math dataset?
> Qwen-Math is a finetuned version of Qwen on math-specific data. Our theory section (Sec 2) shows that training on a given task tends to increase module–data alignment for that specific (model, dataset) pair. Intuitively, for a model that has been fine-tuned on math, the corresponding math dataset (GSM8K in our experiments) activates modules in a more structured and task-aligned way. This is precisely what NFN is designed to capture. Hence, it is natural to observe:
> - Higher NFN scores for Qwen-Math + GSM8K compared to base Qwen + GSM8K.
> - Higher NFN scores for Qwen-Coder + HumanEval (code) compared to base Qwen + HumanEval.
>
> We have made this reasoning explicit in the paragraph "Specialized models show higher NFN scores" in section 3, emphasizing that these empirical patterns are consistent with the theoretical link between training and increased alignment.
>
> ### 2. Baslines and rationale
> **a) Role of PLoP$^{-1}$**: we would like to emphasize that **PLoP$^{-1}$** (inverse PLoP) is used as a **sanity check**, not as a competing method. Its purpose is to show that deliberately choosing module types with the highest NFN scores (i.e., the “most aligned” in our framework) hurts performance, thereby demonstrating that PLoP is capturing meaningful signal rather than noise. We explicitly label PLoP$^{-1}$ as a “sanity check variant” in the experiments section.
>
> **b) Baselines used in our paper and rationale**: We belive there is a misunderstanding on this point. To our knowledge, there are currently no LoRA-specific module-type selection methods in the literature beyond heuristic patterns suggested in prior work, which we include in this paper. This is also reflected in recent practitioner discussions (e.g., Thinking Machines' blogpost on LoRA fine-tuning "LoRA without Regrets") that highlight the lack of principled module-selection strategies. In our paper, we make the rationale for our baselines explicit:
> * ATTN: LoRA on attention modules (Query, Key, Value), as recommended in the original LoRA paper.
> * MLP: LoRA on MLP modules (Down_proj, Up_proj, Gate_proj), as suggested in subsequent empirical studies.
> * ALL: LoRA on all candidate module types.
>
> These correspond to the default configurations used in practice by many practitioners and open-source libraries, which is why we view them as the most relevant baselines for our setting. We added these details in the experiments section.
>
> **c) Additional baselines in the revision:** In response to the reviewer’s suggestions, we have added:
> - Random selection baseline: We select module types uniformly at random (averaged over 4 runs).
> - Base model performance: We report the base model performance alongside all LoRA configurations.
>
> We explicitly discuss these baselines in the experiments section and highlight where PLoP improves over both the standard heuristics (ATTN, MLP, ALL) and random selection.

---

> ### Author Response · Authors · 2025-11-20
> **Authors Response [2/2]**
>
> **d) Oracle baseline and full finetuning:**
>
> - *Oracle baseline*: An “oracle” baseline, defined as the best module-type configuration found via exhaustive search, is computationally prohibitive in realistic settings. For example, even restricting attention to selecting 2 module types out of 7 already requires evaluating 21 distinct module type pair. Each pair requires several runs for learning rate tuning, and this cost grows rapidly with model size. Given that our focus is on practical, scalable methods for large models, we chose not to include such an oracle. We also note that including exhaustive “oracle” baselines is uncommon in the LoRA literature.
> - *Full finetuning*: We believe there might be a misunderstanding on this point. Our goal is to compare LoRA adapter placement strategies, not to compare LoRA against fully updating all parameters. Full finetuning is often infeasible in the scenarios where LoRA is used in practice (restricted compute budget). We emphasize that our contribution is to optimize LoRA placement under a fixed or comparable budget, and therefore we do not believe that full finetuning is relevant to this paper.
> - *Gradient-based methods*: The reviewer also suggested comparing against gradient-based module selection. To our knowledge, there are no established gradient-based methods specifically designed for LoRA module-type selection. We explicitly state this in the paper and note that PLoP is particularly appealing because it requires no gradient computations at all, which makes it well-suited to resource-constrained environments where gradient-based selection may be too costly.
>
> ### 3. PLoP results for SFT (text generation and classification)
>
> - *PLoP results for SFT*: Tables 4 and 5 (SFT for math) show the results of SFT on MetaMathQA for different module selection methods (PLoP, MLP, ATTN, PLoP$^{-1}$, ALL) and varying LoRA ranks $r$ for PLoP. The reason we ablate over $r$ is to match trainable param count between PLoP and the best performing method (not including PLoP). For instance, in Table 1 (Qwen3-0.6B), not including PLoP, we found that MLP is the best method among ATTN, MLP, ALL (taking into account number of trainable params). This was done with $r=64$, which with MLP placement strategy yields $~22M$ trainable parameters. Since PLoP selects "Down_proj", "Up_proj", and "Value" modules, if we use the same $r=64$, we end up with 18.4M trainable param. To ensure fair comparison, we have to match trainable param count, and we picked $r=76$, which yields 21.8M trainable param for PLoP.
>
> **Additional empirical results**: To strengthen the empirical evidence and make the benefits of PLoP more evident, we added the following experiments:
> - *Finetuning on Language datasets*: we finetuning Llama-3.2 1B and 3B models on different language datasets (from AYA dataset) and report the train/test NLL in table 3. The results show that PLoP consistently outperforms other placement strategies.
> - *CommonsenseQA*: we finetune Qwen-2.5 8B and Qwen-2.5 32B models on CommonsenseQA and evaluate answer accuracy. The results show consistent gains from PLoP, and support scalibility of PLoP for large models (32B).
>
> Across these additional empirical results, PLoP consistently outperforms the standard heuristics (ATTN, MLP, ALL) and random selection, at a the *negligible cost* of a single batched forward pass. We summarize these results in Section 4.
>
>
> ### 4. Computational cost of PLoP:
> We expanded the “Compute cost of PLoP” paragraph in Section 3 to directly address the reviewer’s question:
> - Forward-only computation: PLoP requires a single forward pass of the model on a small batch of inputs. In our experiments, we used a batch size of 200.
> - Cost scale: The cost of PLoP is therefore approximately the same as performing standard batched prefill on 200 inputs, which can be merged with the first forward pass of LoRA finetuning. In other, using PLoP adds practically zero overhead.
> - No gradients: PLoP does not require backpropagation or gradient computation at any stage, making it substantially cheaper than any gradient-based selection strategy.
>
> - Sensitivity analysis: to further quantify robustness, we added a new experiment where we vary the batch size from 8 to 1024 and track how often the selected module types change. This is reported in Fig 6 in the revised paper. We find that module-type selection is stable for small batch sizes ($\gtrapprox 64$), which makes PLoP even cheaper.
>
> ### 5. Overall clarity of the Experiments section
> We restructured the section into clearly separated setup and results subsections. We moved detailed descriptions (datasets, learning rate, implementation details) to the appendix and kept only the essential information in the main text.
>
>
> We believe that the additional experiments and clearer presentation significantly strengthen our paper, and we hope these revisions address the reviewer’s main concerns.

---

> ### Author Response · Authors · 2025-11-27
> **Your feedback would be appreciated**
>
> Dear Reviewer vSmF,
>
> Our revised paper was edited to address your main concerns. Please let us know if you have any additional concerns. Your feedback would be much appreciated.

---

### Meta-Review · Area_Chair_pmFU · 2025-12-19

**Summary:**

The article introduces Precise LoRA Placement (PLoP), a lightweight, gradient-free strategy for selecting optimal module types for Low-Rank Adaptation (LoRA) finetuning. By leveraging a theoretical framework based on Normalized Feature Norms (NFN), the authors propose that modules exhibiting low alignment with the target data should be prioritized for adapter placement. The method is validated through experiments on supervised finetuning and reinforcement learning tasks, demonstrating that PLoP often outperforms or matches standard heuristics like placing adapters in all attention or MLP blocks while maintaining a comparable parameter budget.

**Reviewer Concerns:**

During the review process, the committee raised several valid points regarding the initial submission. Primary concerns included the clarity of the NFN definition and the rationale behind the random baseline, the limited scope of the initial experiments (focused heavily on mathematical reasoning and smaller models under 7B parameters), and the lack of comparison against stronger or more diverse baselines. Reviewers also questioned the stability of the NFN rankings with respect to batch size and sequence length, and whether the method could scale to larger architectures. There was also a request to clarify why the method focuses on module types rather than granular layer-wise selection.

Having carefully read the original manuscript and each reviewer's comments and responses at least three times before synthesising this assessment, I note that the authors provided a robust rebuttal. They expanded the empirical evaluation to include 32B parameter models and multilingual datasets (AYA), addressing concerns about scalability and generalization. They also provided sensitivity analyses demonstrating the stability of NFN scores across batch sizes and sequence lengths. The clarification that PLoP is intended to replace ad-hoc heuristics with a compute-efficient alternative—rather than competing with expensive gradient-based search methods—was persuasive. The revisions successfully addressed the majority of the technical and experimental deficiencies identified in the initial round.

**Reviewer Scores:**

Reviewer VHUP explicitly raised their score from 4 to 6 following the inclusion of additional experiments on larger models and diverse datasets, acknowledging the improved value of the work. Reviewers Goqy and VYDu, who initially provided positive assessments, would likely maintain their positive scores given the stability analyses and clarifications provided regarding LoRA variants. Reviewer vSmF, who initially scored a 4, might have improved their assessment had they fully engaged with the inclusion of the random baseline and the logical argument regarding the computational infeasibility of gradient-based selection in this specific context.

The paper presents a practical, empirically validated solution to a common problem in efficient finetuning. The authors have significantly strengthened the manuscript during the rebuttal by adding experiments on 32B parameter models, multilingual datasets, and stability analyses. While the theoretical link between low alignment and adaptation potential is intuitive rather than definitive, the empirical results are compelling. The method's low computational overhead makes it highly relevant for the ICLR community. Therefore, the recommendation is Acceptance.

---

### Decision · Program_Chairs · 2026-01-26

Accept (Poster)